# A framework to measure transit-oriented development around transit nodes: Case study of a mass rapid transit system in Dhaka, Bangladesh

**Md. Anwar Uddin** [1]* , **Md. Shamsul Hoque**[1], **Tahsin Tamanna**[2], **Saima Adiba**[2], **Shah Md. Muniruzzaman**[2‡], **Mohammad Shahriyar Parvez**[2‡]

1 Department of Civil Engineering, Faculty of Civil Engineering, Bangladesh University of Engineering and Technology, Dhaka, Bangladesh, 2 Department of Civil Engineering, Faculty of Civil Engineering, Military Institute of Science and Technology, Dhaka, Bangladesh

☉ These authors contributed equally to this work.
‡ SMM and MSP also contributed equally to this work.
* anwar066.buet13@gmail.com

## Abstract

Transit-oriented development (TOD) is a tool that aids in achieving sustainable urban development. It promotes economic, environmental, and social sustainability by integrating land use and transportation planning. Many researchers have investigated mass rapid transit (MRT) station regions for TOD in developed cities. However, in a developing city such as Dhaka, measuring node-based TOD (TOD index) during MRT construction has been disregarded in planning future land use. Furthermore, no prior research on quantitative TOD measurement in Dhaka exists. As a result, we developed a framework for both quantitative and spatial node-based TOD measurement based on the four Ds (density, diversity, destination accessibility, and design) of the TOD concept. With 17 stations under construction, MRT 6 was selected as our study area. The TOD index was measured by nine indicators based on the four criteria (4Ds), spatially in the geographic information system (GIS). After calculating the indicators, the TOD index for each station's 800m buffer was estimated using the spatial multi-criteria analysis (SMCA). A sensitivity analysis of four TOD scenarios was performed to check the model's robustness. Additionally, a heatmap of the TOD index for MRT 6 was created for informed planning and policymaking. Furthermore, statistically significant hotspots (both Getis Org Gi* and Anselen Local Moran Statistics) and hotspot clusters were identified. Finally, we illustrate the station-based ranking based on the maximum TOD score. In addition, a detailed spider-web of nine indicators for 17 stations depicts sustainable TOD planning. However, regarding density and diversity, sustainable development and (re)development policies should be implemented not only for MRT 6 but for all Dhaka's TOD regions.

**Data Availability Statement:** All relevant data are within the manuscript and its Supporting Information files.

**Funding:** The authors received no specific funding for this work.

**Competing interests:** The authors have declared that no competing interests exist.

## Introduction

Transit-Oriented Development (TOD) has been regarded as a planning strategy that focuses on coupling urban development and transportation around transit nodes while making environment-friendly modes more convenient and appealing for commuting [1]. TOD is also globally acceptable for addressing land use and transportation-related issues [2]. Interestingly, Calthorpe (1993) [3] referred to TOD as a "Pedestrian Pocket," "Compact Neighborhood Development," or simply an "Urban Village." A few years later, Newman and Kenworthy (1996) [4] proposed a transit city with the characteristics of consolidated high-density, mixed-use development around transit terminals. Despite numerous concepts for TOD developed by different authors over time, the main goal is to reduce motorized journeys, increase the share of non-motorized travel, and decrease travel distances with aggravated vehicle occupancy levels. However, in the TOD concept, the transportation network is the backbone. As a result, prioritizing transit provides additional benefits that we hope to receive from TOD design. For example, according to Liang et al. (2022) [5], bus priority on the road decreases traffic congestion, resulting in increased fuel efficiency and environmental preservation. In addition, a dedicated transit lane with an optimized signalized environment increases traffic circulation efficiency for other connecting modes. Moreover, location of transit node plays an important role on traffic circulation and commuter satisfaction. Passengers reap the most significant benefits when transit stops are strategically placed before and after the signalized intersections [6].

The advantages of TOD include urban sustainability, reducing auto-dependent growth, well-planned and designed communities, more accessible public transit with convenience in walking and bicycling, and reducing auto utilization, obesity and other adverse health impacts. According to Singh et al. (2017) [7], TOD should be achieved by eight indicators of density, diversity, pedestrian-friendly urban design, regional development supporting TOD, higher ridership, user-friendly transit system, enhanced connectivity for more accessibility, public transport for longer commutes with additional parking supplies. So, these indicators should be considered to capture the benefits mentioned earlier before implementing any TOD project. Moreover, TOD should be measured quantitatively around the transit nodes before implementing the projects in the real world. Evans and Pratt (2007) [8] highlighted that to evaluate the efficacy of TOD plans accurately, areas must also be assessed for their "TODness," a quantitative measurement of TOD known as the TOD index. We can refer the term TODness as the degree of TOD. Zhou et al. (2019) [9] gave an elaborative definition of TODness. According to him, the magnitude to which the existing conditions of TOD sites fulfil accepted TOD principles frequently entailing significant expenditure and attention. The TOD principles include mixed and dense land use, accessibility, walking and cycling amenities, and compact development with pedestrian-friendly design. His definition also supports the idea of TODness given by Papa and Bertolini (2015) [10] and Singh et al. (2017) [7].

Some well-established studies on TOD measurement were conducted by Z. Li et al. (2019) [11], Motieyan and Mesgari (2018) [12], Singh et al. (2012, 2014, 2017) [7, 13, 14], Teklemariam and Shen (2020) [15] for developed cities. However, very few studies are available for developing cities. Such two studies on TOD measurement by Teklemariam & Shen (2020) [15] and Sulistyaningrum & Sumabrata, (2018) [16] are worth mentioning. Nevertheless, these studies are not well established. Five mass rapid transit (MRT) lines and two bus rapid transit (BRT) lines have been proposed for Dhaka, with MRT 6 and BRT 7 currently under construction [17]. Moreover, ten MRT/BRT hubs and four multi-modal hubs with high TOD potential have been proposed [18]. The missing link between these transit developments could be rediscovered with the prospect of TODness (TOD index). However, no TOD study has been conducted for Dhaka.

We worked on the MRT 6 in Dhaka to demonstrate how the TOD index may be used in node-wise TOD planning. The primary objectives of this paper are to measure the nodal TOD index by developing a framework for emerging cities, and to plan for TOD improvements within walking distance of the station areas based on the results. For nodal TOD index calculation, we use geographic information platforms to consider land use, demographics, and road network characteristics. Then, we identified proper TOD indicators concerning a developing city. Then we aggregated the indicators to get TOD criteria. Finally, we used a spatial multi-criteria analysis (SMCA) to measure TOD index.

The contribution of this research work is the uniqueness of the method because it uses spatial platforms with subjectively defined multi-criteria analysis, and introduces a more established model for developing cities. Furthermore, no TOD index reference value is available for development or comparison of TOD model for Dhaka. Most importantly, MRT 6 is still in development, so no studoy of measuring TOD index has been conducted before construction. As a result, this study could serve as a foundation for TOD planning and a framework for measuring TODness for Dhaka's mega transit projects.

In the following part, we review several literatures to find relevant indicators and criteria for nodal TOD measurement. Then, we establish the appropriate methodology for measuring the TOD index in the context of a developing metropolis. The next section analyzes the identified indicators while doing TOD index's sensitivity, heatmap, hotspot analysis. The following section then elaborates on the findings and recommends station-specific planning solutions. Finally, we conclude this work.

## Literature review

This literature review section focuses on identifying the proper methodology for determining TOD in the context of a developing country. Firstly, we introduce TOD as a sustainable development tool. Then, a decent idea about TOD indicators is revealed based on TOD's sustainability criteria and aims. Finally, a framework of measuring TOD in a spatial platform as better applicability for TOD calculation, TOD index measurement methodology is introduced.

Calthorpe (1993) [3] first introduced TOD as a dilemma to standard auto-dependent development. Since WWII, US cities have become increasingly auto-oriented due to rising automobile ownership, highway development, and dispersed transportation lines. A decade of vehicle reliance has resulted in traffic congestion, noise, health issues, and diminished social life [19–22]. New urbanism and innovative growth policies are reviving public transit in US cities and suburbs [23]. However, TOD is increasingly recognized internationally in regions of intensified urbanization and increased traffic congestion [24]. Therefore, assessing TOD is very critical for planning and policymaking.

One TOD development aim is to use low to high density, walkable neighborhoods with frequent transit stops. The advantages of TOD are access to public transit, fewer car emissions, and enhanced quality of life [7]. However, growing private car ownership is one of the most apparent trends today, and it is a significant issue for many communities. The environmental problems created by the considerable car ownership surges are crucial, yet the increasing number of automobiles may also generate some traffic and non-environmental issues [25]. Moreover, a lack of land use courtesies with massive auto-dependent development does planning and constructing public transportation networks challenging in developing nations. Therefore, land use and transportation planning must be combined to speed up sustainable urban development choices [26].

As sustainable urban is one of the primary focuses of this research, we have to gain pieces of knowledge about sustainable development. "Sustainable development" or "Sustainability" can

be traced to a conference held in Montreal in 1969 by the United Nations, described as finding the highest quality of life for all people and their surroundings concerning the natural resources and the environment [27]. However, a new definition emerged at that time, like "our shared future" on the Brundtland hypothesis. Concerning sustainable development, mass transportation is quintessential because it inflates the expanse of activity and gives more latitude to the individual [28]. Therefore, there is a need to balance people's freedom of movement and resource utilization. Mobility allows them to access additional economic options, promoting increased exorbitance. One possible solution is to make company resources more accessible to the public without reducing automobile use. The three sustainable development aims of conservation, usage, and promotion will boost the economy and keep the world in check. Using technology and pricing, including land use, and decreasing cost promotions are often imminent for sustainable transportation practices [29]. Each plan integrates economics, environment, housing, and transportation, while more system overlaps.

So, addressing the issues related to achieving sustainability, the solution is transit-oriented development. People congregate when jobs and housing are close, and transportation is easily accessible. Applications like these can help achieve sustained growth. Also, it should be kept in mind that TOD and sustainable development are compatible [30]. According to Zuidgeest and Maarseveen (2000) [31], new facilities must be built for sustainable growth. Nevertheless, public transit networks meet citizens' mobility, accessibility, and security needs. Moreover, TOD introduces sustainable development insight and sustainable mobility options. However, one of the most critical concerns on sustainability by TOD has been identified by critics recently is TOD induced gentrification But, Padeiro et al. (2019) [32] argued about that and asserted that the evidence for the gentrification by TOD is ambiguous and not conclusive. So, we can introduce TOD as the most reliable sustainable tool for urban and transport development. Here, Li and Lai's (2009) [33] aggrandization of the fundamental concepts of 3D with sustainable development strategy: density, diversity, and design is portrayed in Fig 1.

However, with more countries beginning to implement TOD projects, it has become clear that the results may be reasonably variable, revealing that a project's success would depend on a wide range of factors, trends, and complicated interrelationships between them [1]. So,

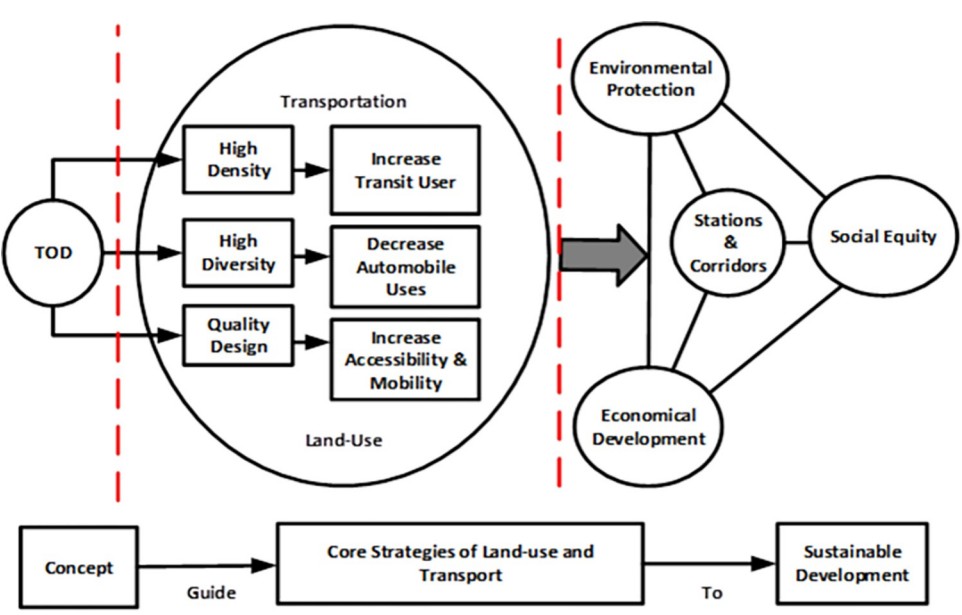

**Fig 1. TOD as a sustainable development.** Source: [33].

measuring the efficacy of complex TOD projects is crucial. Various factors to measure TOD have been discovered over time, such as The 3Ds (density, diversity, and design) were identified by Calthorpe (1993) [3], Evans and Pratt (2007) [8], Renne (2009, 2016) [34, 35], and Ewing and Cervero (2010) [36] as critical factors for TOD measurement. Cervero and Kockelman's (1997) [37] study also used the three indicators to assess TOD. However, they were unable to assess TOD. Renne and Wells (2005) [38] and Evans and Pratt (2007) [8] identified three indicators of TOD economic development. However, Bertolini (1999) [39] emphasized the critical importance of economic indicators before them. As a result, to provide a valid assessment of a TOD, a measure of the 3Ds and economic progress must be included. Finally, Evans et al. (2007) [8] pioneered the "TOD index" to evaluate a practical TOD project. They took both travel and land use into account. It then added a fourth dimension to Cervero and Kockelman's (1997) [37] three-dimensional model, which was used to assess the feasibility of a TOD. On the other hand, Singh et al. (2017) [7] developed eight standards based on the eight rules, which include population density, land use diversity, quality of walking and cycling routes, job creation, number of passengers during peak and off-peak hours, good accessibility of the public transportation system, location of the node, and parking near the node. Moreover, Liu et al. (2020) [2] introduced a new concept named Corridor TOD (CTOD), which expands on the typical nodal TOD idea and practice by capturing corridor level interactions between individual TOD nodes and incorporating economic, social, and environmental variables alongside physical planner/design aspects. Whereas, Niu et al. (2021) [40] introduced Green TOD (GTOD) concept by coupling green urbanism theories and the 5D framework to establish a technique for evaluating the GTOD built environment, based on density, diversity, design, destination and distance.

According to Belzer and Autler (2002) [41], TOD usually emphasizes the physical form or appearance of the project rather than the purpose for which it is designed. Therefore, they proposed the Station Level Application Criteria (SLAC). They believed they could demonstrate TOD's full potential on a regional scale. Nonetheless, Dittmar and Poticha (2004) [42] used SLAC based on some performance criteria. They emphasized gathering quantitative data on each outcome to calculate these criteria. Nevertheless, these criteria are data-intensive and frequently impossible to obtain. As a result, Renne (2007) [43] advocated two strategies: the Regional Performance Approach (RPA) and the Community Performance Approach (CPA). Using these methods, he attempted to evaluate the design viability of TOD. However, Singh et al. (2014) [14] proposed two indices for measuring TOD: the actual TOD index and the potential TOD index. The actual TOD index, by definition, examines how much current TOD exists near transportation facilities, specifically within 800m of the station's buffer. On the other hand, the potential TOD index measures accessibility on a regional scale. The actual TOD index for each station's buffer has been determined by this study (800 m). Renne (2016) [35] stressed the significance of stakeholders' perspectives on TOD measurement. Moreover, He emphasized the Multi-Criteria Analysis (MCA) guidelines for calculating the TOD. Ibraeva et al. (2020) [1] also discovered obstacles in bringing together multidisciplinary stakeholders in one frame to create user-friendly decision support systems.

After discussing various TOD measurement methods, spatially explicit analyses of tool-oriented design are necessary. City form should also be visualized for public participation in the planning process [44]. Therefore, TOD planning requires a spatial analytical platform (SAP). SAP offers numerous benefits. First, spatial analyses consisting of maps and plans are necessary when negotiating TOD. Second, TOD should include catchment areas for the neighborhood's various transportation systems. Third, these areas must be within a 250–500 m radius of the transit station or stop. Finally, non-geographical indicators can be viewed with a Geographic Information System (GIS) application [13].

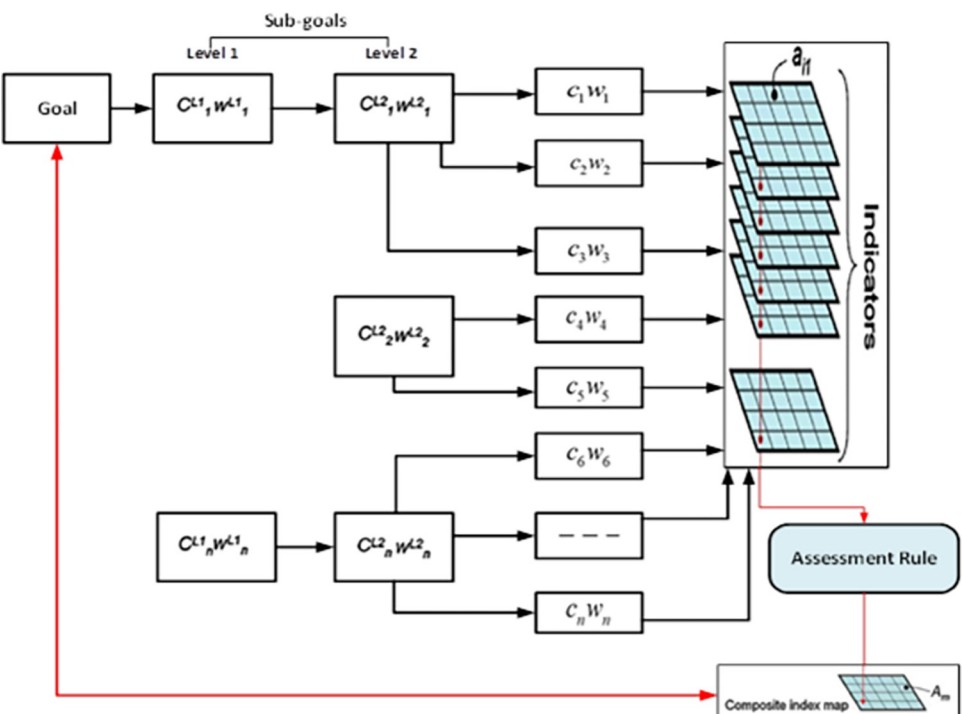

**Fig 2. Algorithm to assemble composite index maps from multi-criteria analysis of geographic data.** Source: [50].

Nevertheless, the decision-making framework is essential for this research. In this framework, people make decisions when they want to act or not based on prescriptive and normative theory. The perceptive theory focuses on decision execution, while normative theory focuses on what decisions should be made. Both approaches use math to simulate and analyze the decision-making environment [45]. Therefore, Beukes et al. (2011) [46] adopted a multi-criteria decision-making (MCDM) methodology to identify the ideal solution. Usually, MCDM differs from portfolio selection by using qualities to identify alternatives [47]. So, GIS analysis is strongly recommended because of its strength in automating, analyzing, and administering geographic data. In addition, GIS is a form of data integration designed to improve decision-making, according to Cowen (1998) [48]. Likewise, most spatial decisions involve location. A previous study found that choosing the best option from a set of options and criteria is part of decision making. So, GIS can help with alternatives and decision criteria. A spatial multi-criteria evaluation strategy using GIS and MCDM is called spatial multi-criteria evaluation (SMCE) [49]. Fig 2 illustrates the steps linked to performing the SMCA. First, SMCA's primary goal is divided into subgoals, criteria, and indicators. Then, each indicator is linked to these weights once measured. From these weights and values, a composite value is calculated. This number dissolves the target.

## Methodology

The components that drive transit-oriented development must be assessed and aggregated to measure TOD near transit nodes. Moreover, other spatial and non-spatial variables can be employed to generate unique solutions. Thus, the index should contain both spatial and non-spatial indicators. However, as a potentiality check was conducted based on existing land use around the stations, which have not been operational yet, spatial variables were our fundamental concerns.

## Identifying TOD indicators

As previously discussed in the literature review, the following rules have been used for urban development and transit characteristics as they relate to an area's overall level of TOD:

**Rule 1.** The growth of urban densities is the key to fostering new TOD.

**Rule 2.** Placement of various land uses and the diversity of the land uses would result in a dynamic environment from a single location.

**Rule 3.** More equitable and robust connectivity with walking and cycling and a greater frequency of service improve the development of a TOD.

**Rule 4.** Open space with ample parking for bikes and cars would empower more individuals to travel via public transit for longer trips.

Four rules have been matched with four criteria (4Ds). As a result, indicators were used to evaluate each of the four criteria. Furthermore, several well-known and widely used indicator variables used in several successful TOD case studies have been identified by Balz and Schrijnen (2009) [51], Lindau et al. (2010) [52], and Newman (2009) [53]. Table 1 depicts the criteria and indicators with reference and index.

## Calculating TOD indicators

To determine the TOD index, the TOD indicators have been calculated first. As 17 stations have been chosen for node-based TOD index calculation, a buffer radius of 800 m has been used around each station. The reason behind choosing an 800m buffer is that even while most of the neighborhood types currently have densities and mixed-land uses equivalent to other TOD cities in Asia, those within the standard zone (800 m) from MRT have considerable potential for TOD planning [54]. After evaluating vector data, raster maps of all indicators were generated. Fig 3 depicts the vector maps of Pallabi station's land use, road network, and building footprint representative of 17 stations (GIS shapefiles of all other 16 stations' land use, road network, and building footprint for Fig 3 have been provided in S1 File). To determine the TOD for each grid cell (GC), the buffer area (BA) has been subdivided into GCs, ensuring that the GC size is neither too large nor too small through an index calculation. Different grid tessellations (GT) of 100x100m, 200x200m, 300x300m, and 500x500m have been investigated for this study. Singh et al. (2015) [55] examined a 1000 $km^2$ region using a 300 x 300 m grid. In this study, the buffer area of each station is approximately 2.011 $km^2$; a 300x300m buffer grid

**Table 1. Criteria and indicators for gauging TOD index.**

| Criteria | Indicators | Reference, Index |
|---|---|---|
| Density | Population density | Number of persons /$km^2$ |
| | Commercial density | Number of commercial enterprises/ $km^2$ |
| | Employment density | Number of employees/ $km^2$ |
| Diversity of land use | Land use diversity (Mix percent) | Measure of entropy |
| Destination accessibility | Land use mixedness | Mixed index (Mixedness of residential land use with other land uses) |
| | Length of walkable/cyclable paths | km |
| | Intersection density | Number of intersections/$km^2$ |
| Design | Parking utilization | Surface parking spaces |
| | Open/green spaces | Green/open areas, amenities |

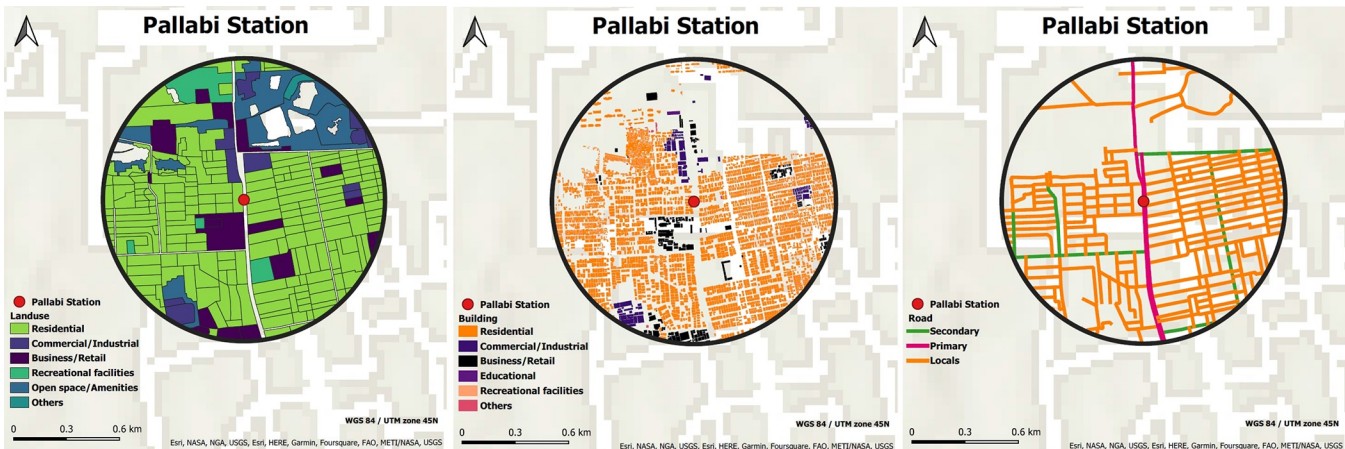

**Fig 3.** Vector maps of Pallabi station's buffer: (a) Land use (Source: https://datacatalog.worldbank.org/search/dataset/0039604, CC BY- 4.0 and https://apps.nationalmap.gov/viewer), (b) Building footprint (Source: https://apps.nationalmap.gov/viewer), and (c) Road network (Source: https://datacatalog.worldbank.org/search/dataset/0042062, CC BY- 4.0 and https://apps.nationalmap.gov/viewer).

cell (BGC) with approximately 224 BGCs has been used in each station buffer (SB). The selected GT is more suitable for this research study, and according to Singh et al. (2015) [55], it is computational efficiency.

Furthermore, unlike traditional TOD developments, specific TOD areas have been covered by a 500-meter walking distance. The National Transit Oriented Development (TOD) Policy defines the influence zone of TOD as 500–800 m if the station spacing is approximately 1 km. If the station spacing is less than 1 km, the influence zone becomes 500 m because of overlapping [56]. Moreover, according to TOD standard framework by Institute for Transit Development & Policy (ITDP), within a 5 km radius, rapid transit connections with frequent bus stops should be accessible at around 500 m. However, as our study is for a developing south Asian city, we take the guideline of Cochin and Mumbai city metro by considering various south Asian city's guidelines. The influence area for these two cities has been adopted as a 500 m buffer for Both cities [57]. As both cities are densely populated, like Dhaka, we take their guideline for our analysis radius. As a result, the following measures were implemented to allow for a 500 m radius analysis window around the cell centroid (CC). First, the TOD Service Cell (TSC) area was used instead of the BGC area to calculate the indicators. The TSC is the influence area of each BGC that serves a specific cell within a 500 m buffer. The TSC covers 0.79 km$^2$. Then, the calculated TSC indicator value was then delineated as the corresponding BGC value. As a result, the index of that specific BGC was calculated using all the indicators measured from each BGC value.

**Population density (PD).** PD was estimated using population data from the BBS (2015). The population per km$^2$ of each neighborhood has been calculated using population data. For the 100x100m BGC TSC, data apportionment was used to split each community's population and apply it to the TSC area. Building footprint and land use data have also been used for apportionment based on the percentage of land occupied by residential building footprints in the community's geographic area. However, a reference value of PD is necessary for ideal TOD planning. According to the Florida Department of Transportation (FDOT) [58], PD of min 135 persons/acre, 100–145 persons/acre, and 80–135 persons/acre is ideal for urban core, urban general, and suburban areas respectively considering TOD.

**Commercial density (CD).** The building footprints of commercial structures were used for data apportionment, and CD was calculated as the number of commercial enterprises per

km$^2$. The FDOT [58] recommends a commercial spaces-housing ratio of 10 commercial spaces per dwelling unit in urban cores and 5 commercial spaces per dwelling unit in other metropolitan regions.

**Employment density (ED).**   ED was calculated by dividing the number of employees per km$^2$ by the same method used to calculate the other two density measures. In this case, non-residential building footprints were used to compute data apportionments. Nevertheless, FDOT recommends the ED value of min 1000 jobs/acre, 190 to 250 jobs/acre and 35 to 80 jobs/acre for urban core, urban general and suburban areas respectively [58].

**Land use diversity (LUD).**   The LUD aims to estimate the variety of land uses within a specific region and how they are geographically distributed within that land area [55]. It also includes a variety of lands used in various types of dense metropolitan areas. The level of variety in this study was measured using the term "entropy," which was previously modified by Eck and Koomen (2008) [59] and diversity has been depicted in the following 2 equations.

$$LU_d(i) = \frac{-\sum_i Q_{lu_i} \times ln(Q_{lu_i})}{ln(n)} \tag{1}$$

Where,

$$Q_{lu_i} = \frac{S_{lu_i}}{S_i} \tag{2}$$

$LU_d(i)$ = land use diversity in the analysis area $i$

$lu_i$ = land use class (1,2,. . . . ., n) within analysis area $i$

$Q_{lu_i}$ = The share of land usage by application in the analysis area $i$

$S_{lu_i}$ = Specific land use area throughout the area being examined $i$

$S_i$ = The entire realm of analysis $i$

A score of 0 on diversity shows the absence of land use, whereas a value of 1 represents the balanced land use. Only the benefits of urban land, including residential, commercial/industrial, business/retail, educational, recreational, and others, have been considered during this computation.

**Land use mixedness (LUM).**   LUM considers the accessibility of nearby destinations. A disparity exists between various land uses and diverse land uses. Trip destinations, including business and leisure, can be reached on foot with enough nearby mixed-use developments. The LUM was calculated using Zhang and Guindon's (2006) [60] findings using the following equation.

$$MI(i) = \frac{\sum_j L_0}{\sum_j L_r + L_0} \tag{3}$$

Where,

$MI(i)$ = Mixedness Index

$L_0$ = Non-residential land uses for each TSC j

$L_r$ = Residential land uses for each TSC j

MI values may range from 0 to 1, MI value of 0.5 suggesting a 50/50 split between residential and other land uses. However, FDOT recommends 20/80, 50/50, 70/30 split between residential and non-residential land uses for urban core, urban general and suburban areas respectively [58].

**Intersection density (ID).**   Intersections and route length measure walkability. High-intersection regions have shorter walking distances. So, the number of junctions per km$^2$ from the road network data has been used to calculate the ID in the analysis area.

**Length of walkable/cyclable paths (LWC).**   The number of pedestrian and cycle-accessible roads near each transit station determines walkable areas. Typically, distances are measured in meters. A road network's design assumes vehicles will travel at a moderate speed. These roads were excluded from the road network data because they pose a risk to pedestrians and cyclists. Schlossberg and Brown (2004) [44] implemented a reclassification system for the road network, estimating walkable and cyclable paths by determining connectable road segment lengths. Using this method, the study reclassified roads based on walkability and cyclability.

**Parking utilization (PU).**   Dhaka's parking should be used more efficiently. Large parking lots near transit hubs may lead to excessive driving, so optimizing parking spaces is critical. Conversely, small parking lots can be counterproductive. Monitoring parking occupancy can identify stations with limited parking and decide whether to add retailers or bike parking (if required). However, one study by Ewing et al., 2021 [61] showed a dilemma among shared, residential, and rented parking spaces in which 51.2%–84.0% of parking spaces were filled during the peak demand near the transit hubs. So, concerning the importance of PU, this study has used GIS and OSM to investigate surface parking (SP) around transit nodes. However, FDOT recommends surface parking of maximum 10%, 70% and 80% of total area for urban core, urban general and suburban areas respectively [58].

**Open/Green spaces (OGS).**   Designing optimized amenities and open/public spaces around the transit hubs is critical to creating a liveable environment. According to Appleyard et al. (2019) [62], individuals' and society's overall well-being are positively correlated with stations having higher livability ratings. A pleasing design also impacts the city's overall quality and helps improve public transit hubs. However, GTOD, introduced by Niu et al. (2021) [40], expounds on environmental and ecological components emphasizing urban green space (part of green urbanism) and sustainability. Therefore, sustainable criteria OGS has been extracted from existing land use data.

## Calculation of TOD index

All spatial indicators related to TOD's 4D concept (Density, Diversity, Destination Accessibility, and Design) were used in this study. The Quantum GIS (QGIS) 3.22 and Aeronautical Reconnaissance Coverage Map (ArcMap) 10.3 tools were used to calculate and aggregate the indicators. For the SMCA analysis, TerrSet 2020's IDRISI GIS Analysis was used. The index was created using both vector and raster data formats. When calculating indicators, vector data has some advantages over raster data. Such as:

- Highways and parcels have irregular vector shapes. TOD uses topology and network analysis. As a result, indicators can be analyzed as vector data, which is easier and faster. [63, 64].

- Neighborhoods and district divisions collect urban planning data. Vector space is more compatible than raster space due to socioeconomic homogeneity [12].

However, raster is preferred over vector for calculating the TOD index. IDRISI's TOD index requires 100x100 raster input. SMCA raster output was used to create heatmaps. This paper combines vector and raster data to make it easier and faster to figure out the TOD index.

Notably, the aggregation of indicators requires the same unit. However, our SMCA indicators have different units. So, all indicators have been standardized using the maximum standardization technique (MST) to give them the same unit. In addition, MST has applied a 0–1 gradient to all values. Usually, the gradient shows how each indicator affects the TOD index.

However, a consolidation of 'benefit' and 'cost' has been used to account for mixed land use, allowing the index to behave as a 'benefit' until 0.5. When the value exceeds 0.5, it becomes a "cost" for the index and lowers TOD [14].

The adopted indicators have been balanced after standardization. Then each criterion was weighted. Typically, each weight indicates the criterion's importance for the TOD. So, the criteria and indicators have been arranged in order of their importance. However, not every indicator will be linked to TOD rationalization in any project. As stakeholders have different perceptions, the weightings reflect that. According to MCA principles, the indicator weighting is computed before the TOD index. Planners, academics, and community members are often weighed in [7]. Therefore, academicians from different universities in Dhaka and personnel from DMTCL have been selected as stakeholders in this research.

Respondents were asked to prioritize the TOD index indicators. As a result, respondents assigned a weight to a criterion indicator. Next, Reilly's (2002) [65] numbers were combined using a "Borda Count" process. The "importance" options were then sorted according to "importance." Following order management, the participant's first option received a single score of "n," their second option received a score of "n minus 1," and so on. Finally, the ranks of all respondents were converted to a score for each "candidate," which was then added together. The candidate with the highest rank received the highest score, while the candidate with the lowest rank received the lowest. The final weights were computed using the rank-sum method and a modification of Singh et al. (2015) [55] (Table A in S1 Table). This modified rank-sum approach has been depicted the following equation.

$$W_k = \frac{n+1-k}{\sum_i^n (n+1-i)} \tag{4}$$

Where,
$W_k$ = normalized weight for the criterion with rank k
$n$ = total number of criteria in the set
$i$ = index of summation that takes the value from 1 to n

Finally, the weights were loaded into IDRISI, and TOD index maps were generated. Model robustness is vital for SMCA, so sensitivity analysis with some alternative scenarios was performed. Fig 4 depicts a detailed framework for calculating the TOD index using SMCA.

## Study area

The MRT 6, which is currently undergoing implementation in Dhaka, has been selected to measure the Node-based TOD index. MRT 6 already has seventeen planned stations (nodes). Therefore, the "area of analysis" should be defined to measure TOD near each transit node accurately. Furthermore, TOD is based on creating walkable neighborhoods [3]. Therefore, we reviewed additional literature to determine the appropriate walking distance. It varies between 250 and 800 meters based on location and demographics. Nevertheless, Singh et al. (2017) [7] state that a 10-minute walking radius from the station hub should be used to measure TODness. Therefore, we have decided on an 800 m (10-minute walk) buffer between each station (Fig 5) (GIS shapefiles of 17 stations for Fig 5 have been provided in S2 File).

## Data collection

To conduct this research, accurate and sincere data collection was performed. In addition, secondary data sources consisting primarily of precise and trustworthy information were utilized. Table 2 depicts the secondary data sources with their corresponding parameters for analysis.

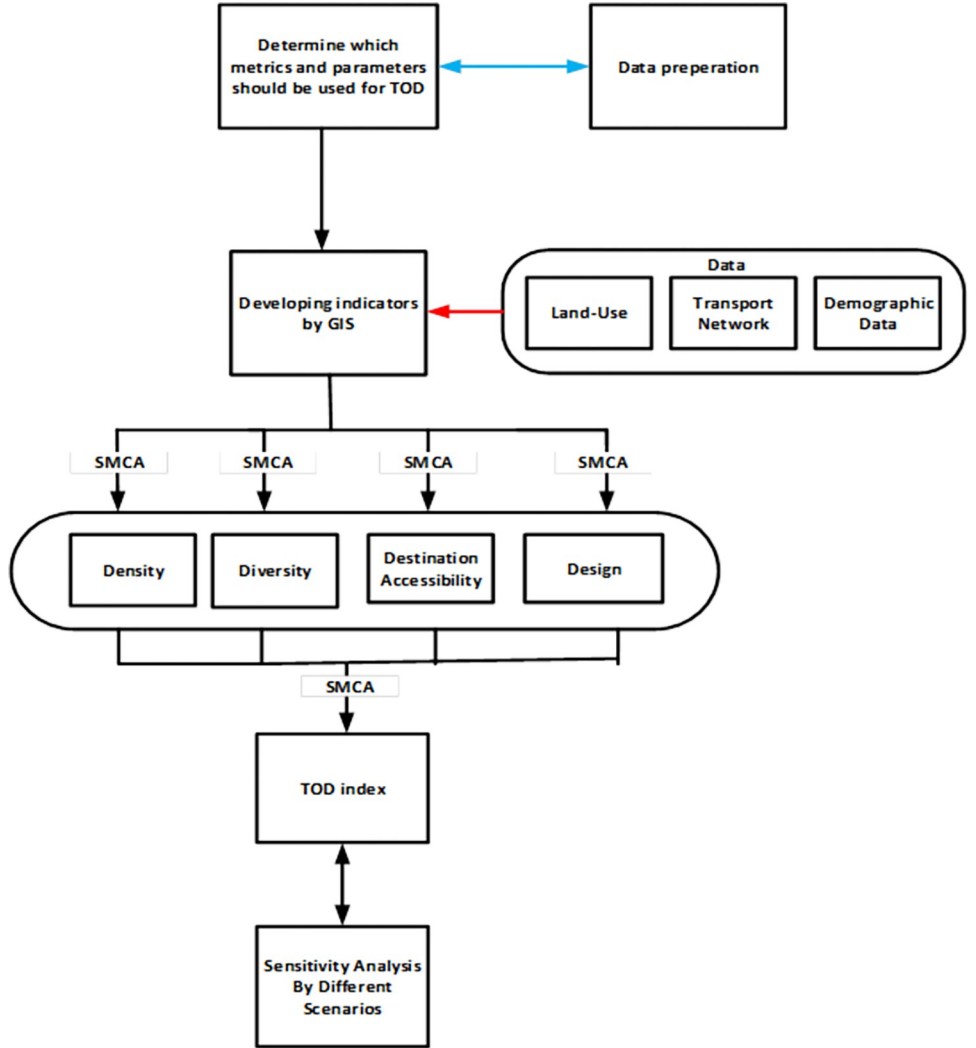

**Fig 4. SMCA analysis framework for TOD index calculation.**

## Result analysis and findings

Firstly, a correlation analysis of the indicators has been conducted in this section. The correlation analysis will give insights into the relationship among the indicators, which will help planners and policymakers with proper TOD planning. Secondly, a sensitivity analysis has been executed. Typically, different scenarios are developed with different weightage in a sensitivity analysis. The reason for performing this analysis is that the SMCA is widely regarded as an effective tool for resolving spatial choice quandaries. However, its use of probable outcomes has been called into question [66]. In addition, some uncertainty is associated with the weights provided by those who lack extensive experience and knowledge [67]. So, from different scenarios with weightage, we can know the criteria of TOD changes to confirm how robust the model is. Then, a heatmap of the TOD index has been created, which will help the policymakers and urban-transport planners identify the high TOD zones for MRT 6. As a result, there will be alternative options for them to choose the situation better and implement it for a greater purpose. Moreover, a hotspot with spatial autocorrelation has been performed in this

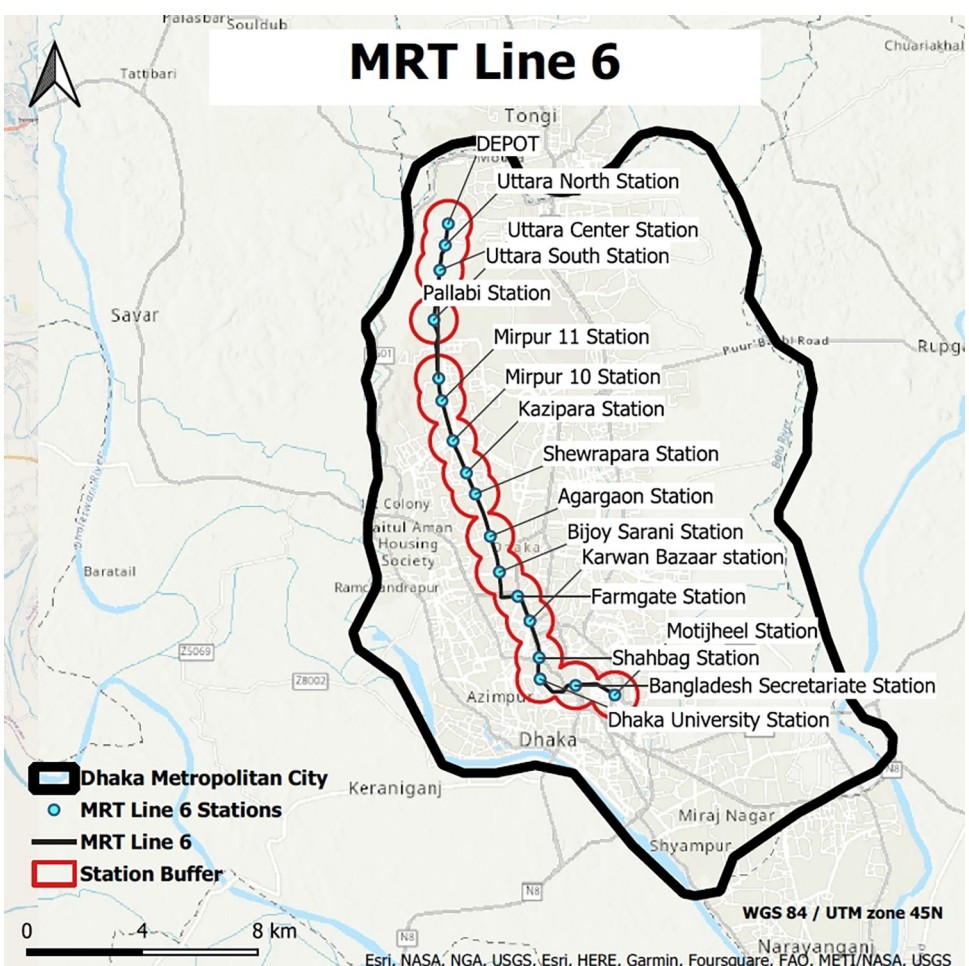

**Fig 5. MRT line 6 with 17 stations (source: Map is prepared by authors using base map from https://apps. nationalmap.gov/viewer).** Figure is similar but not identical to the original image and is therefore for representative purposes only.

analysis for preliminary planning. These hotspot maps will be valuable for proper land use policymaking for the buffer of individual stations. Finally, the ranking of the stations based on the max TOD index has been performed to help policymakers to decide which buffer areas need more concern for TOD improvements.

**Table 2. Sources of data.**

| Data | Type of data | Sources of data |
|---|---|---|
| Demographic data | Statistical data | Bangladesh Bureau of Statistics (BBS-2015) |
| Building footprint data | Vector data | Dhaka South City Corporation (DSCC), Dhaka North City Corporation (DNCC), Open Street Map (OSM) |
| Land use data | Vector data | Detailed Area Plan (DAP) of Rajdhani Unnayan Kartripakkha (RAJUK), World Bank Land Use Data (2017) |
| Transportation road network | Vector data | World Bank Road Network Data (2017) |
| Other data | Map | Google map, Dhaka Mass Transit Company Limited (DMTCL) Website |

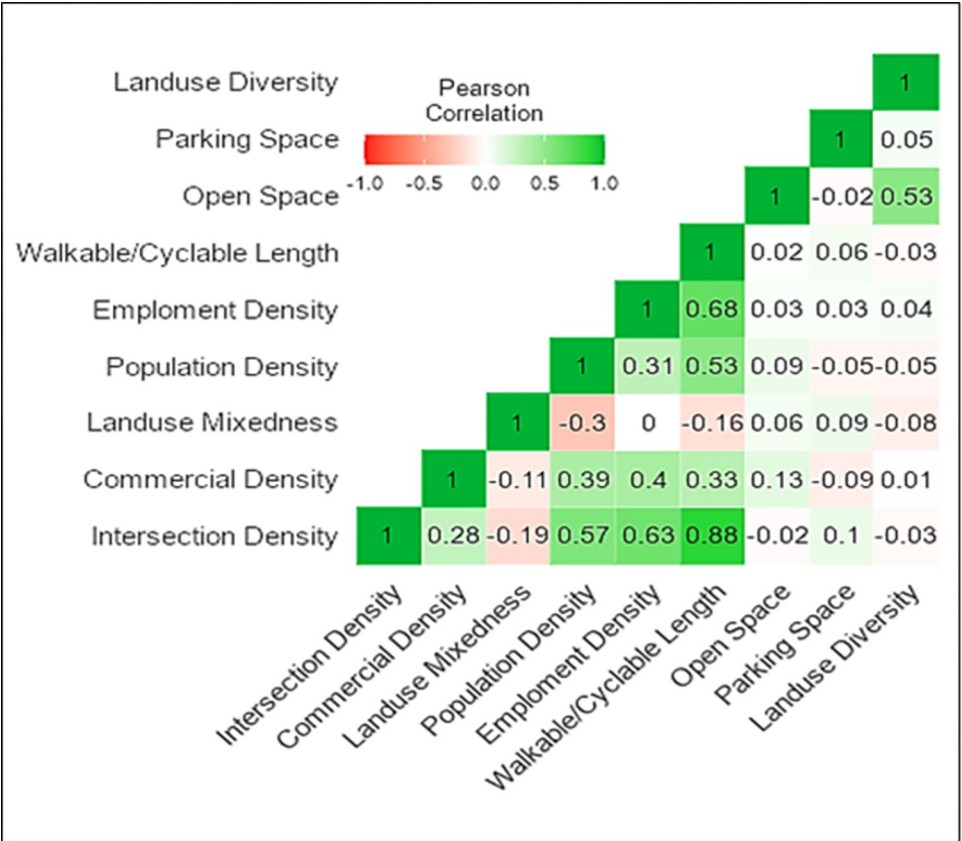

**Fig 6. Correlation heatmap of indicators.**

## Correlation analysis of indicators

Fig 6 depicts the partial correlation of nine indicators. There is a strong positive correlation between intersection density and the length of walking or cycling paths. The correlation value is 0.88, indicating that more road intersections provide more accessible walking and cycling paths. Intersecting density is also strongly related to population and employment density. It implies that improved accessibility will result in more jobs and, thus, more people will live nearby. Moreover, the TOD design requirements link well with the diversity indicator. However, an intense negative correlation between land use mixedness and population density has been noticed, suggesting that more people living near TOD stations will have a less balanced land use mix. Residential land use should be balanced with other land uses. For the people serviced by the buffer area of each TOD station, a mix of commercial, industrial, and other land uses with residential land uses will provide a more balanced environment. The relationship between indicators can help policymakers consider transportation and land use in TOD development.

## Sensitivity analysis of TOD index

Standardized values of nine indicators (85[th] percentile values of all BGCs) have been represented in Table 3 to calculate the TOD index. These values can be used for vivid inferences of the TOD index with the indicators and recommendations for sustainable TOD policies. Moreover, standardization of indicators is a necessary part of the SMCA analysis.

**Table 3. Criteria and indicators' standardized values for 17 stations' buffers.**

| Station Name | Density | | | Diversity | Destination accessibility | | | Design | |
|---|---|---|---|---|---|---|---|---|---|
| | Population density | Commercial density | Employment density | Land use diversity | Land use mixedness | Length of walkable/cyclable paths | Intersection density | Parking utilization | Open/ green spaces |
| Depot | 0.78 | 0.79 | 0.87 | 0.74 | 0.85 | 0.88 | 0.75 | 0.00 | 0.17 |
| Uttara North | 0.75 | 0.86 | 0.70 | 0.54 | 1.00 | 0.83 | 0.65 | 0.00 | 0.15 |
| Uttara Center | 0.75 | 0.00 | 0.76 | 1.00 | 1.00 | 0.76 | 0.63 | 0.00 | 1.00 |
| Uttara South | 0.79 | 0.00 | 0.80 | 0.59 | 1.00 | 0.68 | 0.80 | 0.00 | 0.11 |
| Pallabi | 0.85 | 0.74 | 0.81 | 1.00 | 0.71 | 0.78 | 0.75 | 0.00 | 1.00 |
| Mirpur 11 | 0.84 | 1.00 | 0.77 | 0.78 | 1.00 | 0.85 | 0.84 | 0.00 | 0.75 |
| Mirpur 10 | 0.80 | 0.87 | 0.82 | 0.85 | 1.00 | 0.86 | 0.81 | 0.00 | 0.84 |
| Kazipara | 0.88 | 0.92 | 0.75 | 0.84 | 0.68 | 0.91 | 0.87 | 0.00 | 1.00 |
| Shewrapara | 0.82 | 0.00 | 0.76 | 0.79 | 0.90 | 0.85 | 0.86 | 0.00 | 1.00 |
| Agargaon | 0.78 | 0.83 | 0.75 | 0.93 | 1.00 | 0.81 | 0.76 | 0.00 | 1.00 |
| Bijoy Sarani | 0.73 | 0.98 | 0.76 | 0.89 | 1.00 | 0.76 | 0.83 | 1.00 | 1.00 |
| Farmgate | 0.77 | 0.79 | 0.80 | 0.67 | 0.91 | 0.86 | 0.79 | 0.68 | 0.34 |
| Karwan Bazar | 0.78 | 0.80 | 0.81 | 0.61 | 0.85 | 0.87 | 0.71 | 0.00 | 0.44 |
| Shahbag | 0.82 | 1.00 | 0.87 | 0.87 | 1.00 | 0.79 | 0.78 | 0.31 | 1.00 |
| DU | 0.00 | 0.00 | 0.79 | 1.00 | 1.00 | 0.78 | 0.84 | 1.00 | 1.00 |
| BD Secretariat | 0.90 | 0.71 | 0.79 | 0.82 | 1.00 | 0.79 | 0.78 | 1.00 | 1.00 |
| Motijheel | 0.86 | 0.70 | 0.83 | 0.78 | 0.62 | 0.88 | 0.74 | 0.00 | 1.00 |

From Table 3, it has been found that the population density for all the stations ranges from 0 to 0.90. Moreover, for commercial and employment density, the standardized value ranges from 0 to 0.92 and 0.87. However, the maximum and minimum values for diversity are 0.54 and 1.0. However, maximum values for accessibility indicators such as mixedness, length of walkable/cyclable paths, and intersection density are 1.0, 0.91, and 0.87, respectively. Interestingly, there is zero parking space for almost all stations except Bijoy Sarani, Farmgate, Shahbag, BD Secretariat, and DU stations. The maximum value for open space is 1.0 for the maximum number of stations on MRT 6, and the minimum value is 0.11 for Uttara South station.

For sensitivity analysis, different weights were applied to each scenario to calculate the TOD index, with a ranking of the criteria (Table 4). It should be noted that no reference weightage value has yet been developed for Dhaka's TOD measurement. Furthermore, the exercise of assigning weightage is subjective. The weightage assigned to the analysis can change over time or in different environments. So, changing the weightage within a range [67] is beneficial for using sensitivity analysis.

**Table 4. Criteria's weightage with index based on different scenarios.**

| Criteria | Base scenario | | Scenario 1 | | Scenario 2 | | Scenario 3 | |
|---|---|---|---|---|---|---|---|---|
| | Rank | Weights | Rank | Weights | Rank | Weights | Rank | Weights |
| Density | 1 | 0.35 | 1 | 0.40 | 1 | 0.40 | 2 | 0.30 |
| Diversity of land use | 1 | 0.35 | 2 | 0.25 | 2 | 0.30 | 1 | 0.40 |
| Destination accessibility | 2 | 0.20 | 2 | 0.25 | 3 | 0.20 | 3 | 0.20 |
| Design | 3 | 0.10 | 3 | 0.10 | 4 | 0.10 | 4 | 0.10 |

However, from Table 4, it has been found that scenario highest weightage is given to density and land use diversity (0.35) and minimum weightage is given to design (0.10) for base. Similarly, for scenario 2 and 3, maximum and minimum weightage is given to same as base scenario. But maximum weightage value is 0.40 in both cases. For scenario 3, maximum and minimum weightage is given to land use diversity (0.40) and design (0.1) respectively. It is noted that the weightage is given based on different ranking by modified rank-sum approach.

The descriptive statistic of the TOD index of four scenarios for the 17 stations of the MRT line was calculated (Table 5). A comparative study among the stations can be inferred for the MRT 6 with the maximum and lowest TODness. As the reference value of the TOD index is absent for Dhaka, the index can be compared for a similar group of criteria, similar purposes, and the same input parameters. As this study was conducted for the first time and no reference value is available for Dhaka, our TOD index can be a reference source for better TOD planning in the future.

From Table 5, the maximum TOD index value for all 17 stations was observed within a reasonable range. We can observe that the highest deviation of the max value of the TOD index has ranged from 0.441 to 0.569. However, the max value varies for the stations. The max value ranges from 0.521 to 0.871 for the base scenario. However, for scenarios 1, 2, and 3, the maximum value range is 0.441 to 0.859, 0.473 to 872, and 0.561 to 0.871, respectively. Interestingly, the min value of the TOD index for all stations is 0. Moreover, the highest deviation of the mean value has ranged from 0.333 to 0.379, which is within the allowable limit. Nevertheless, the range of the mean value for the base scenario, scenarios 1, 2 and 3 is 0.172 to 0.429, 0.169 to 0.420, 0.175 to 0.256, 0.163 to 0.427, and 0.183 to 0.43. The range of the max, min, and mean values designated for the stations is identical for all scenarios. It is also to be noted that the stations' overall ranking for all four scenarios has remained identical, indicating that the sensitivity analysis does not affect the results. Last but not least, the sensitivity analysis shows that the framework for figuring out index values is strong [55].

## Heatmap of TOD index

A pristine view of TODness in different stations of MRT 6 has been observed in the developed heat maps (Fig 7) from the raster data of the TOD index (GIS shapefiles of TOD index heatmap of 17 stations for Fig 7 have been provided in S2 File). The analysis shows that the buffers of multiple stations at some zones highly overlap. These are the high stimulus zones of TODness. Depot, Pallabi, Mirpur 11, Kazipara, Farmgate, Karwan Bazar, and Shahbag stations show potential zones for TOD. The statistically significant hotspots have also been identified to better understand buffer-based land use planning potentials.

## Hotspot analysis of TOD index

First, the spatial correlation of the hotspot values of the TOD index must be calculated before hotspot analysis. The hotspot analysis will be robust if they are statistically significant. ArcGIS 10.3 has been used to calculate spatial autocorrelation.

**Spatial correlation analysis.** In our autocorrelation, the null hypothesis is "the spatial distribution of the dataset is not clustered in nature". However, in Fig 8, it can be observed that the p-value is statistically significant, and the z-score is positive. Thus, the null hypothesis has been rejected. Therefore, the spatial distribution of high and low values of the dataset has been observed as more clustered than expected. So, underlying spatial processes are random [68].

Global Moran's I index has been determined from this spatial statistical analysis. Global Moran's I value is 0.564. This Moran's index value is excellent. They range from -1 to 1. As the value is above 0.5, a good correlation can be inferred between the hotspots of the index [68].

**Table 5. Descriptive statistics of TOD index for different scenarios.**

| | | Depot | Uttara North | Uttara Centre | Uttara South | Pallabi | Mirpur 11 | Mirpur 10 | Kazipara | Shewrapara | Agargaon | Bijoy Sarani | Farmgate | Kawran Bazar | Shahbag | DU | BD Secretariat | Motijheel |
|---|---|---|---|---|---|---|---|---|---|---|---|---|---|---|---|---|---|---|
| Base Scenario | Min | 0.0 | 0.0 | 0.0 | 0.0 | 0.0 | 0.0 | 0.0 | 0.0 | 0.0 | 0.0 | 0.0 | 0.0 | 0.0 | 0.0 | 0.0 | 0.0 | 0.0 |
| | Max | 0.706 | 0.682 | 0.713 | 0.540 | 0.776 | 0.836 | 0.811 | 0.848 | 0.677 | 0.844 | 0.648 | 0.763 | 0.735 | 0.871 | 0.521 | 0.703 | 0.709 |
| | Std. dev | 0.219 | 0.195 | 0.223 | 0.161 | 0.218 | 0251 | 0.233 | 0.236 | 0.197 | 0.253 | 0.221 | 0.227 | 0.218 | 0.238 | 0.189 | 0.234 | 0.215 |
| | Mean | 0.370 | 0.243 | 0.364 | 0.234 | 0.354 | 0.429 | 0.394 | 0.362 | 0.247 | 0.212 | 0.207 | 0.211 | 0.228 | 0.214 | 0.172 | 0.201 | 0.200 |
| Scenario 1 | Min | 0.0 | 0.0 | 0.0 | 0.0 | 0.0 | 0.0 | 0.0 | 0.0 | 0.0 | 0.0 | 0.0 | 0.0 | 0.0 | 0.0 | 0.0 | 0.0 | 0.0 |
| | Max | 0.711 | 0.679 | 0.678 | 0.541 | 0.751 | 0.828 | 0.812 | 0.829 | 0.642 | 0.825 | 0.691 | 0.741 | 0.741 | 0.859 | 0.441 | 0.679 | 0.668 |
| | Std. dev | 0.218 | 0.205 | 0.209 | 0.162 | 0.218 | 0.250 | 0.226 | 0.232 | 0.193 | 0.246 | 0.208 | 0.224 | 0.217 | 0.231 | 0.169 | 0.224 | 0.207 |
| | Mean | 0.357 | 0.246 | 0.333 | 0.220 | 0.352 | 0.420 | 0.374 | 0.347 | 0.246 | 0.200 | 0.191 | 0.207 | 0.225 | 0.206 | 0.160 | 0.190 | 0.191 |
| Scenario 2 | Min | 0.0 | 0.0 | 0.0 | 0.0 | 0.0 | 0.0 | 0.0 | 0.0 | 0.0 | 0.0 | 0.0 | 0.0 | 0.0 | 0.0 | 0.0 | 0.0 | 0.0 |
| | Max | 0.718 | 0.689 | 0.687 | 0.542 | 0.769 | 0.828 | 0.817 | 0.842 | 0.654 | 0.833 | 0.674 | 0.748 | 0.744 | 0.872 | 0.473 | 0.687 | 0.684 |
| | Std. dev | 0.223 | 0.202 | 0.216 | 0.159 | 0.218 | 0.250 | 0.231 | 0.238 | 0.193 | 0.256 | 0.217 | 0.226 | 0.220 | 0.234 | 0.175 | 0.228 | 0.216 |
| | Mean | 0.369 | 0.244 | 0.348 | 0.226 | 0.358 | 0.427 | 0.388 | 0.362 | 0.249 | 0.208 | 0.201 | 0.212 | 0.229 | 0.209 | 0.161 | 0.196 | 0.201 |
| Scenario 3 | Min | 0.0 | 0.0 | 0.0 | 0.0 | 0.0 | 0.0 | 0.0 | 0.0 | 0.0 | 0.0 | 0.0 | 0.0 | 0.0 | 0.0 | 0.0 | 0.0 | 0.0 |
| | Max | 0.699 | 0.675 | 0.739 | 0.538 | 0.784 | 0.840 | 0.806 | 0.853 | 0.699 | 0.854 | 0.622 | 0.778 | 0.727 | 0.871 | 0.569 | 0.718 | 0.734 |
| | Std. dev | 0.217 | 0.190 | 0.230 | 0.164 | 0.218 | 0.252 | 0.236 | 0.234 | 0.202 | 0.255 | 0.227 | 0.228 | 0.217 | 0.243 | 0.203 | 0.240 | 0.215 |
| | Mean | 0.371 | 0.241 | 0.379 | 0.243 | 0.350 | 0.431 | 0.399 | 0.361 | 0.245 | 0.215 | 0.213 | 0.211 | 0.227 | 0.219 | 0.183 | 0.206 | 0.199 |

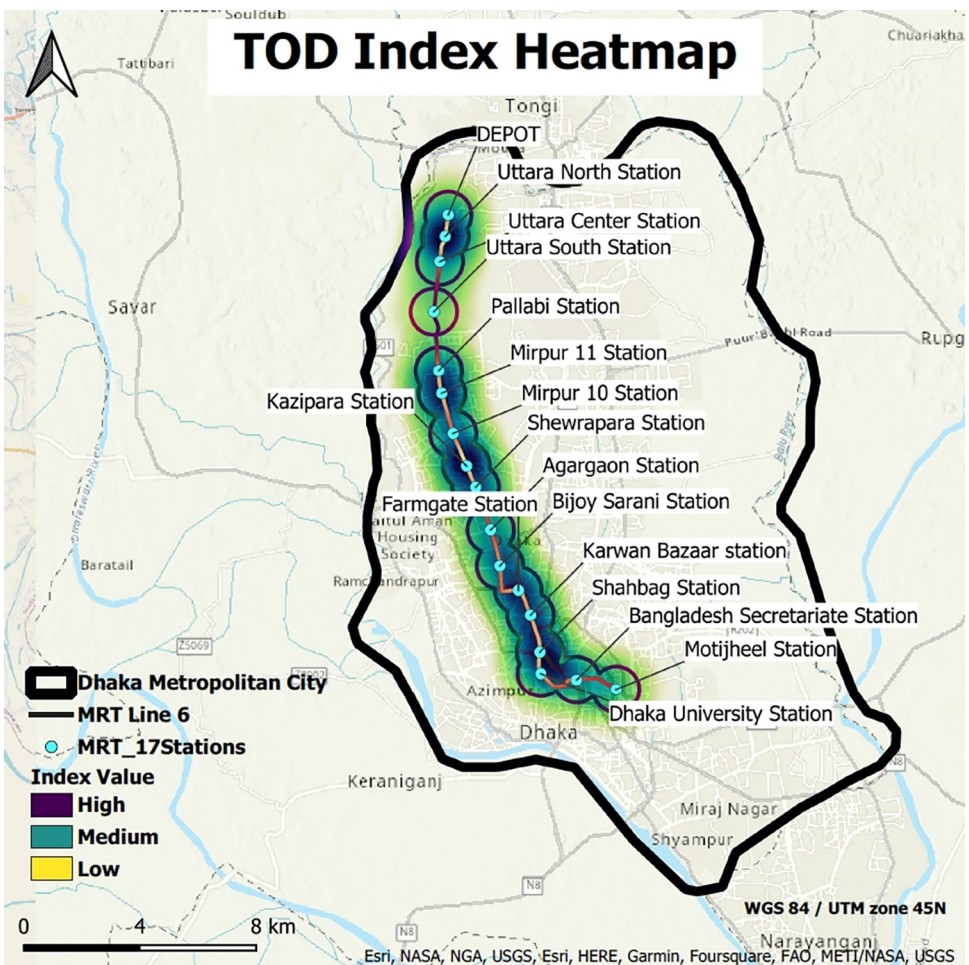

**Fig 7. Heatmap of TOD index (source: Map is prepared by authors using base map from https://apps. nationalmap.gov/viewer).** Figure is similar but not identical to the original image and is therefore for representative purposes only.

**Hotspots of TOD index.** Both Getis Ord Gi* and Anselin Local Moran's statistical analyses have been performed in this research. Consequently, visual identification has been made from the hotspot map (Fig 9) for the most TOD-influenced stations of MRT 6 (GIS shapefiles of TOD index hotspots of 17 stations for Fig 9 have been provided in S3 File).

From Fig 9, it has been found that Mirpur 11, 10, Kazipara, and Pallabi stations are significant for both Getis Ord Gi* and Anselin Local Moran's statistical analysis, which have been identified as the hotspot clusters. So, it can be inferred that the buffers of these three stations depict the highest potential TOD hotspots. Also, some portions of the Depot, Farmgate, Shahbag, Agargaon, Karwan Bazar, and Motijheel stations show the potential TOD hotspots. The result of the analysis also supports the developed heatmap.

## Ranking of stations based on max TOD index

TOD score of different stations varies from different scenarios. However, from sensitivity analysis, the fluctuation of the TOD scores shows reasonable limits. As the cell values differ in the same station buffers, the max value of the tod index has been taken for the analysis. The max value represents the individual station's TOD value of the whole buffer. The TOD index max

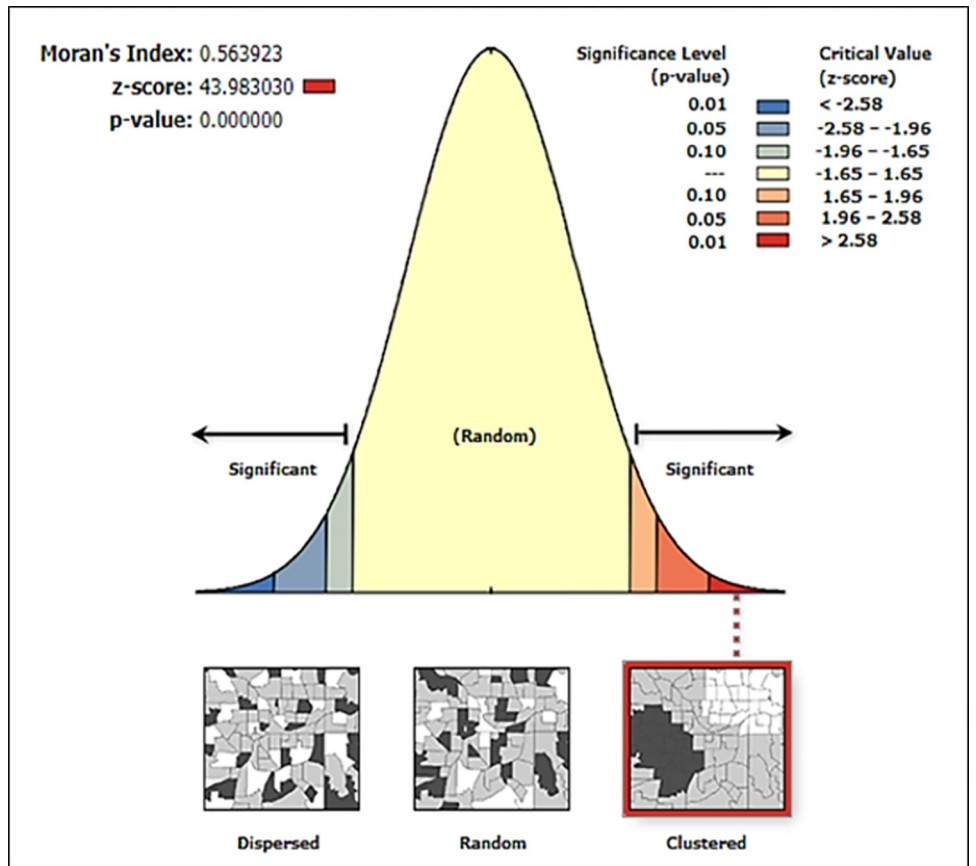

**Fig 8. Spatial autocorrelation.**

value map has been portrayed in Fig 10 (GIS shapefiles of max TOD of 17 stations for Fig 10 have been provided in S2 File).

A detailed ranking of the stations (Fig 11) has been made based on the max TOD index. Based on the ranking, the stations have been classified into three categories. Among the 17 nodal hubs of the MRT 6, five stations have been identified as top-ranked, seven as medium-ranked, and five as low-ranked.

It has been observed that the top-ranked stations for all scenarios are Shahbag, Mirpur 11, Agargaon, Kazipara, and Mirpur 10 stations (Fig 11). From hotspot analysis (Fig 9), it can be augmented that three hotspot clusters, Mirpur 11, 10, and Kazipara, are the buffers of top-ranked stations resulting from the max TOD index ranking system. Pallabi, Farmgate, Kawran Bazar, Bijoy Sarani, Uttara Center, Uttara North, and Depot stations have been identified as medium-ranked stations. Based on TOD index scores, it can be concluded that BD Secretariat, Motijheel, Shewrapara, Uttar South, and DU stations are the low-scoring stations that need further improvement.

Spider-webs of indicators of all the stations based on TOD scores have been depicted in Fig 12 for more detailed TOD planning.

From Fig 12, it has been observed that the maximum value of LUM, CD, and OGS has been observed for the Shahbag station, which is the highest scoring station. Whereas Mirpur 10 and 11 depict almost the same indicator value except for CD, Mirpur 11 shows more value for CD. The maximum potential indicators for the Agargaon station are OGS, LUM, and LUD.

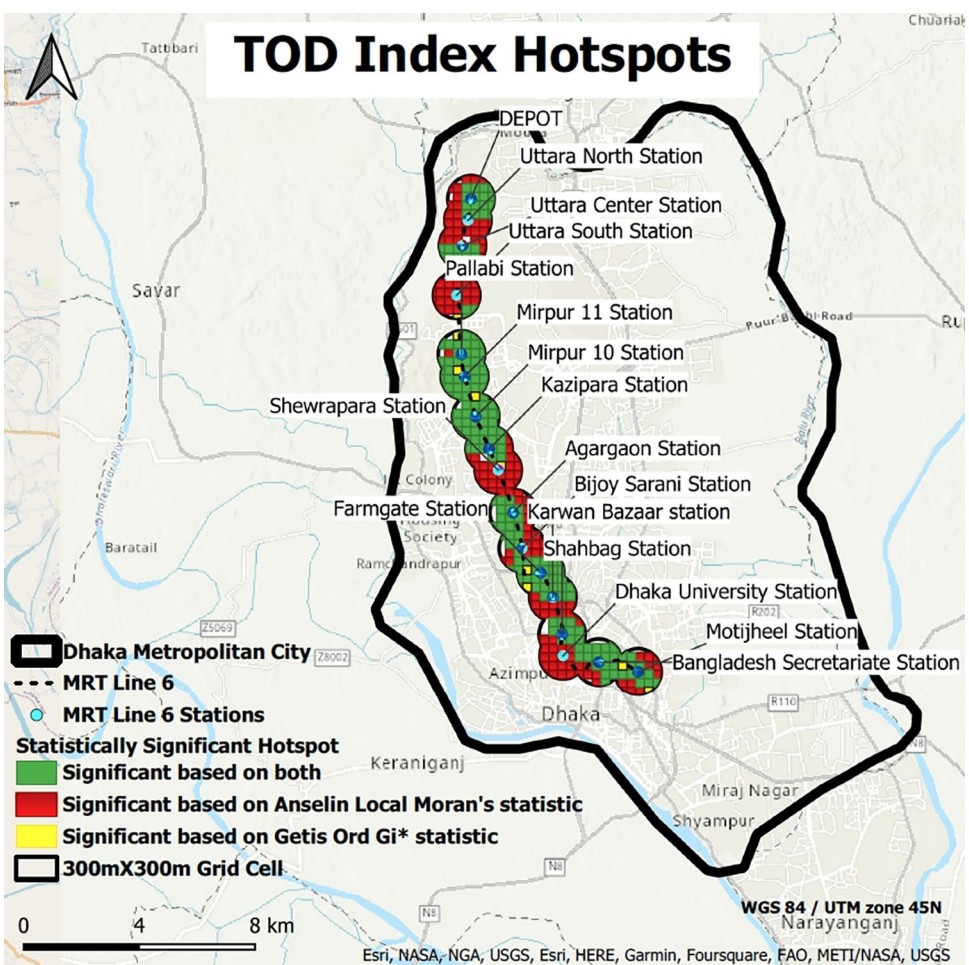

**Fig 9. TOD index hotspots (source: Map is prepared by authors using base map from https://apps.nationalmap. gov/viewer).** Figure is similar but not identical to the original image and is therefore for representative purposes only.

However, CD, OGS, and LWC are suitable indicators for the Kazipara station. However, as the lowest scoring stations, DU, Uttara South, Shewrapara, Motijheel, and BD Secretariat stations have different indicator values. The highest indicators for DU stations are OGS and PU. The Uttara south, Shewrapara, and BD Secretariat stations show the highest value for LUM. Moreover, the Motijheel station offers the highest OGS value. However, medium scoring stations show variable indicator values. Therefore, individual station-based indicator analysis for the medium scoring stations will not be significant to discuss. So, based on the findings, the highest and lowest scoring stations are more important than the medium-scoring stations.

## Discussions

TOD index values should be compared, considering local conditions to begin the TOD planning process for a city. The significant finding of this research is that the transit stations that serve many transportation modalities have high TOD index ratings. Most TOD serving stations are Shahbag, Mirpur 11, Agargaon, Kazipara, and Mirpur 10, with high index scores from analyses of all scenarios. Max score ranges from 0.81 to 0.87. These stations are the beating hearts of Dhaka, showing the most significant potential. These max TOD degrees have been found in Dhaka's urban core stations. Typically, there is a consistent spatial pattern to the

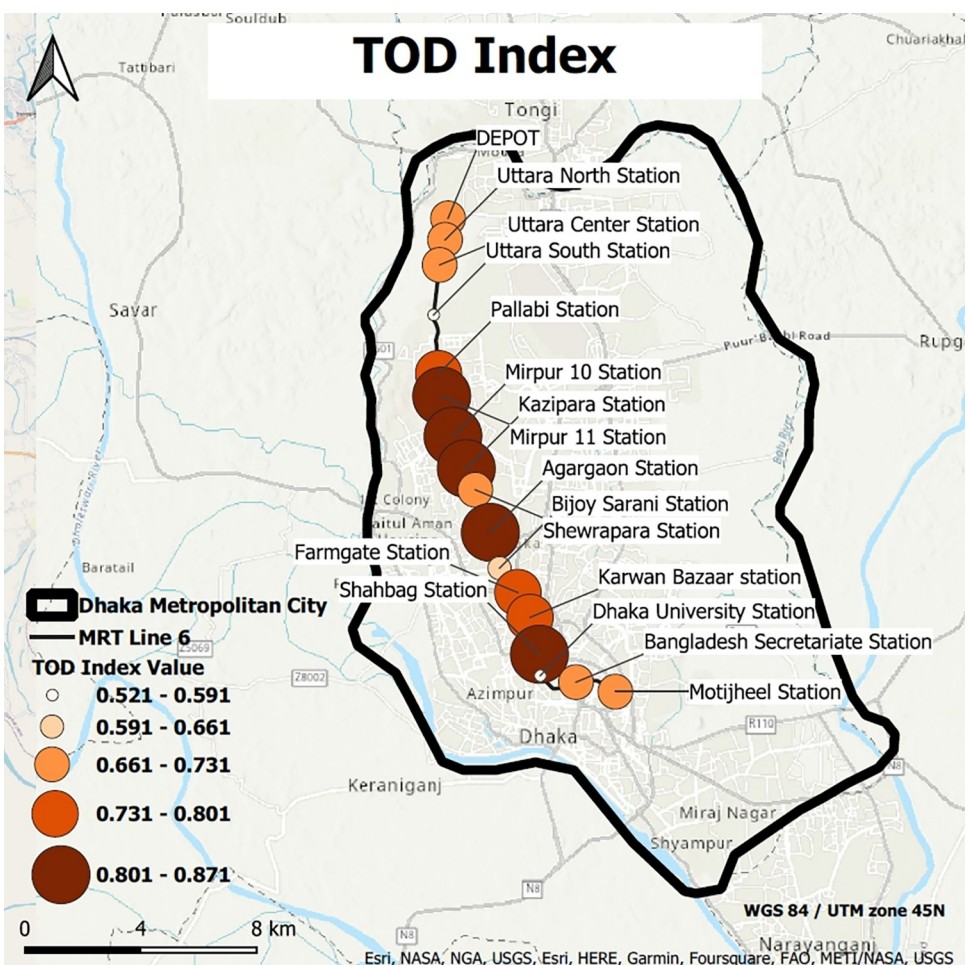

**Fig 10. Max TOD index (source: Map is prepared by authors using base map from https://apps.nationalmap.gov/viewer).** Figure is similar but not identical to the original image and is therefore for representative purposes only.

TOD degree in terms of its drop from the core to the periphery, regardless of city size [69]. However, to better plan, each station's buffer area, the indicators' values should also be considered. In addition, scores of the criteria such as land use diversity and density are greatly affected by the locations of the nodes, which means that the cost of current public transport services at these nodes is significantly higher than it is for other stations.

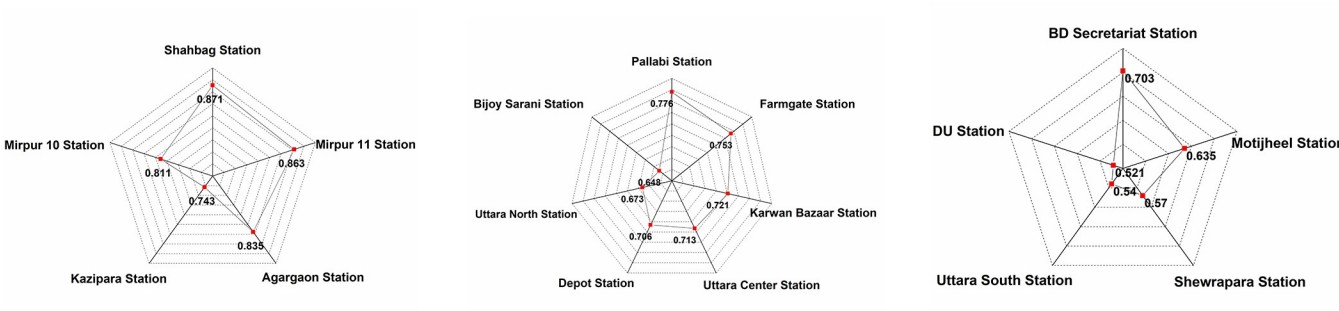

**Fig 11.** Ranking of stations based on max TOD index for (a) Top-ranked, (b) Medium-ranked, (c) Low-ranked stations.

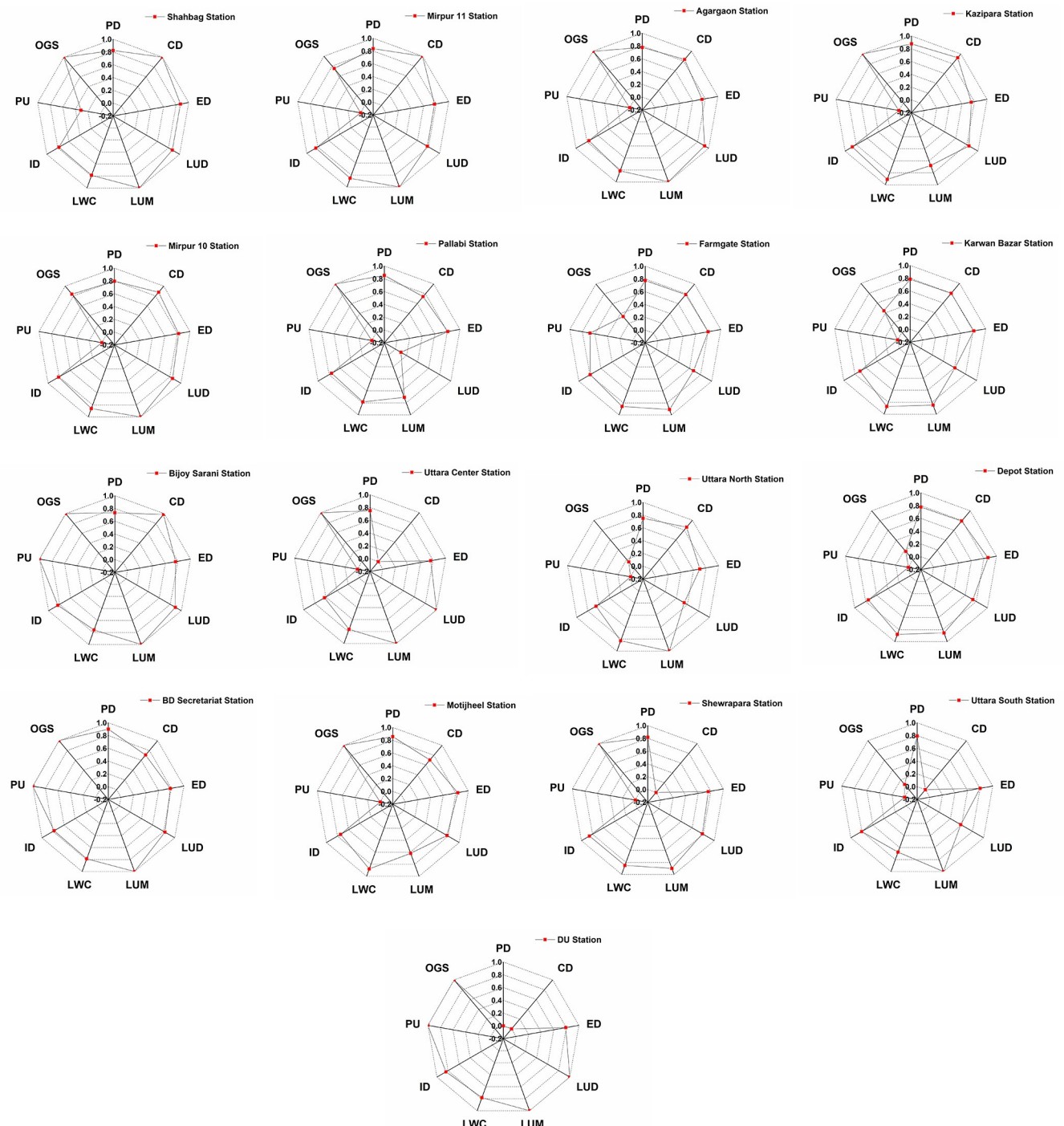

**Fig 12.** Web diagrams of TOD indicators of 17 stations–(a) Shahbag, (b) Mirpur 11, (c) Agargaon, (d) Kazipara, (e) Mirpur 10, (f) Pallabi, (g) Farmgate, (h) Karwan Bazar, (i) Bijoy Sarani, (j) Uttara Center, (k) Uttara North, (l) Depot, (m) BD Secretariat, (n) Motijheel, (o) Shewrapara, (p) Uttara South and (q) DU station (Arranged in order based on TOD score).

Sahabag station scoring 0.871 has a relatively high score among all stations, considered one of Dhaka's important transport hubs. The reason behind high values can be the highest level of retail facilities being available in this area in Dhaka. Second suitable TOD scoring (0.836) stations is Mirpur 11. Commercial density (1.00) is also highest for this station. So the reasons of

highest TOD for these stations support the findings of Ganning and Miller (2020) [70] that the TOD Index appears to be the main driver of greater retail activity, with a correlation to a higher level of activity density. Other two suitable stations for TOD identified as Kazipara (0.848) and Agargaon (0.844) stations. For both, OGS indicator is common. LWC, LUM and LUD are also important indicators for these high scoring stations. So LUM, LWC, LUD and OGS are the indicators that should be given more importance for TOD planning in Dhaka. OGS correlates positively with LUD indicating that diversity in land use planning will be suitable for livable neighborhoods for the residents. Another critical indicator LUM shows an intense negative correlation with PD, which suggests that more people living near TOD stations have a less balanced land use mix. Therefore, residential land use should be balanced with other land uses. However, a negative correlation of LUM with design indicators indicates that balanced mixedness in land use promotes more walking, cycling, and a livable environment. So, destination accessibility criteria for the LUM indicator should be improved to get more captivity from TOD development. Moreover, according to Zacharias and Zhao (2018) [71] wide, straight pedestrian routes should be provided to the train stations to improve local accessibility with consideration of design criteria. So, for TOD planning in Dhaka as a developing city, a pedestrian and cyclist-friendly livable community design should be encouraged.

However, the minor scoring station's index must be observed to identify the criteria for sustainable land use planning for TOD. DU (0.521), Uttara South (0.540), Shewrapara (0.677), BD Secretariat (0.703), Motijheel (0.709), and stations are the lowest-scoring stations. These neighborhoods are primarily residential. Not all nearby transportation nodes have parking areas that accommodate vehicles and non-motorized users.

Regarding the environment, including parks and open spaces, not all stations have been able to keep up. The lowest score has been found for the DU station. From indicator analysis, it has been found that less diversity value (0.00) is behind it. Only educational land use has been observed, indicating that diverse land use balanced with residential will be most suitable for TODness.

In Dhaka, the essential criteria of MRT 6 for TOD are density. Therefore, planners and policymakers should emphasize density while considering TOD. The present land use scenario of Dhaka is densely developed. As this TOD index is the actual TOD scenario of Dhaka's current land and transport condition, density is the primary supporting tool for TOD development in Dhaka city, which is the inherent strength of the urban form of Dhaka. Diversity is also essential for TOD development in Dhaka, especially for MRT line 6. However, according to Zaręba et al. (2019) [72], density, location efficacy, and redevelopment strategies are critical for sustainable planning. As density for our study has been more emphasized, redevelopment strategies should be recommended for MRT 6. For proper (re)development policies, according to Song et al., 2021 [73], institutional barriers should be overcome by land value capturing to make an integrated transit and land development. Moreover, Papagiannakis et al. (2021) [74] have also emphasized bypassing financial barriers for TOD. Given the high-density characteristics of these neighborhoods of MRT 6, TOD planning and policy in these areas should take lessons from Asian cities such as Hong Kong, Shanghai, and Seoul [54].

As accessibility is also a well-influencing factor in MRT 6 TOD strategy, accessibility is not in good shape in the adjacent buffer region of the individual MRT 6 station. Hence accessibility should be improved more to get more captivity from TOD development. On the other hand, design criteria influence less considering the present TOD scenario in Dhaka. The main reason is that as TOD hubs are not operational yet, design criteria will affect more towards TODness when more parking space is needed for the stations. Moreover, considering design criteria, to improve local accessibility, wide, straight pedestrian routes should be provided to the train stations [71].

However, we have proposed walking-based TOD; walking has a high positive correlation for MRT 6 with intersection density which supports most of the literature. Nevertheless, Zacharias and Zhao (2018) [71] argued that the density of intersections has a negative effect on walking distance, and the presence of commerce has a positive effect. In our case, both support the walking-based accessibility for the TOD. For MRT 6 buffer areas, a high rate of destination accessibility can occur in high commercial land uses with densified employment. High intersection density also increases accessibility and makes a more preferred walking environment.

Furthermore, in Curitiba, making traffic flow smoother, building and preserving affordable housing, and implementing zoning policies were successful. Zonal policies assist the mix of activities, which benefits the local economy and helps raise density. Moreover, regulating land use and urban planning at the zone level, assisted by zonal policies, encourages more people to use the local metro [71]. However, there is still room for residential and employment density (population and employment density indicators). In addition, pedestrians must be given priority. Moreover, low-cost housing and density should be given more emphasis.

To improve TODness at the node level, some strategies should be taken based on the indicators (Fig 12) that have the most potential for improvement (Table 6).

**Table 6. Indicators with most potentials for improvement.**

| Ranking | Station name (Ordered based on TOD score) | Indicators with the most potential for improvement based on the findings of web-diagram |
|---|---|---|
| **Top ranked stations** | Shahbag Station | Parking utilization, Population density, Intersection density, Length of walkable/cyclable paths |
| | Mirpur 11 Station | Parking utilization, Open/green Spaces, Employment density, Land use diversity |
| | Agargaon Station | Parking utilization, Employment density, Intersection density |
| | Kazipara Station | Parking utilization, Mixedness Index, Employment density |
| | Mirpur 10 Station | Parking utilization, Population density, Employment density, Intersection density |
| **Medium ranked stations** | Pallabi Station | Parking utilization, Land use diversity, Mixedness Index, Commercial density |
| | Farmgate Station | Parking utilization, Open/green spaces, Land use diversity, Population density |
| | Karwan Bazar Station | Parking utilization, Open/green spaces, Intersection density, Land use diversity |
| | Bijoy Sarani Station | Population density, Employment density, Length of walkable/cyclable paths |
| | Uttara North Station | Parking utilization, Open/green spaces, Land use diversity, Intersection density, Population density, Employment density |
| | Uttara Center Station | Parking utilization, Intersection density, Population density, Employment density, Length of walkable/cyclable paths |
| | Depot Station | Parking utilization, Open/green spaces, Intersection density, Land use diversity |
| **Low ranked stations** | BD Secretariat Station | Commercial density, Intersection density, Length of walkable/cyclable paths, Employment density, Land use diversity |
| | Motijheel Station | Parking utilization, Intersection density, Mixedness Index, Commercial density, Land use diversity |
| | Shewrapar Station | Parking utilization, Commercial density, Employment density, Land use diversity |
| | Uttara South Station | Parking utilization, Open/green spaces, Commercial density, Land use diversity, Length of walkable/cyclable paths |
| | DU Station | Population density, Commercial density, Employment density, Length of walkable/cyclable paths |

## Conclusions

This paper has developed a framework for measuring TOD quantitatively (TOD index) around all transit nodes of MRT 6 in Dhaka using SMCA. Identifying useful indicators and criteria for MRT 6, quantitative analysis of them for proper correlation, analyzing them spatially station by station, developing a spatial model of TOD index with sensitivity analysis for the robustness of the model, creating heatmap and hotspots of TOD index, and station-based ranking with node level in detail analysis are the main contributory parts of this paper. This paper has also demonstrated how the TOD index is necessary for sustainable planning in a developing city like Dhaka, where no study on TOD has been conducted yet. The paper also demonstrates how the index compares the transit node areas over the urban form or region. Finally, following the analysis of the TOD index, it can be concluded that in Dhaka, MRT 6 development characteristics encourage TOD development. Planners and legislators should keep density in mind while making TOD decisions. Dhaka is heavily built, which provides a range of land use planning difficulties for the future. Due to the city's high density, Dhaka's land and transit provisions should be optimized in the TOD index. Moreover, the necessity for diversity is a paramount concern to successfully implementing TOD in Dhaka, especially in the Mass Rapid Transit line. Regarding density and diversity, sustainable development and (re)development policies should be applied not only for MRT 6 but also for the Dhaka regions where TOD represents the city's future. However, some node, buffer area, and urban level development strategies can be implemented for long-term TOD planning. For example, the node-level policy includes preparing ahead of time for eki-naka development to prevent interrupting MRT routes and operations. Furthermore, buffer area level rules include modal transportation amenities, safety, comfort, and convenience for pedestrians, transit access, and park and ride. Finally, the incorporation of an official urban plan with the proposed MRT line, assimilation into the Detailed Area Plan (DAP) of the MRT district of Dhaka, and promotion of urban development along the transit corridor of Dhaka can be viable options for urban level development planning. Nevertheless, density zoning, reliable transportation infrastructure that fits new development trends, and land readjustment with acute phase development plans can all be recommended policies for (re)development.

Nevertheless, this paper has some limitations, which can be the steppingstone for the future scope of research. Firstly, station clustering was not done in this study for MRT 6. However, this clustering will give insight into node typology for indicator selection, index measurement, and decision making. So, this opens up another research opportunity for clustering TOD stations of MRT 6, which will improve TOD planning and implementation. Secondly, Once MR 6 stations are operational, non-spatial variables such as passenger information, transit fare, transit frequency, passenger boarding-alighting, station-based parking facilities, and amenities can be added to the model, expanding the research scope. Finally, according to Huang and Wey (2019) [75], ecological variety, available energy regeneration, and a habitable environment should be included in urban planning and design, not merely the sustainability aspect of traditional TOD.

## Supporting information

**S1 Table. Indicator weights with criteria.**
(DOCX)

**S1 File. GIS shapefiles of land use, building footprint and road network of 17 stations.**
(ZIP)

**S2 File. GIS shapefiles for MRT line 6 with 17 stations, heatmap, max TOD index.**
(ZIP)

**S3 File. GIS shapefiles for TOD index hotspots.**
(ZIP)

## Acknowledgments

The authors wish to acknowledge Md. Amin Al Noor for coordination on data collection for this research. We also thank the Editage for the proofreading and for giving proper suggestions for improvement of the abstract part of this paper. Finally, we appreciate anonymous reviewers' comments to improve this paper's quality.

## Author Contributions

**Conceptualization:** Md. Anwar Uddin, Md. Shamsul Hoque.

**Data curation:** Md. Anwar Uddin, Tahsin Tamanna, Saima Adiba.

**Formal analysis:** Md. Anwar Uddin.

**Investigation:** Saima Adiba.

**Methodology:** Md. Anwar Uddin.

**Project administration:** Md. Shamsul Hoque.

**Software:** Md. Anwar Uddin, Tahsin Tamanna, Saima Adiba.

**Supervision:** Md. Shamsul Hoque.

**Writing – original draft:** Md. Anwar Uddin.

**Writing – review & editing:** Md. Anwar Uddin, Shah Md. Muniruzzaman, Mohammad Shahriyar Parvez.

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
