## [Decision Letter · Decision Letter 0]

22 Aug 2022

PONE-D-22-18746A framework to measure transit-oriented development around transit nodes: Case study of a mass rapid transit system in Dhaka, BangladeshPLOS ONE

Dear Dr. Anwar,

Thank you for submitting your manuscript to PLOS ONE. After careful consideration, we feel that it has merit but does not fully meet PLOS ONE’s publication criteria as it currently stands. Therefore, we invite you to submit a revised version of the manuscript that addresses the points raised during the review process.

We look forward to receiving your revised manuscript.

Kind regards,

Sheng Jin

Academic Editor

PLOS ONE

Journal Requirements:

  "NO"

    "NO"

5. Please ensure that you include a title page within your main document. You should list all authors and all affiliations as per our author instructions and clearly indicate the corresponding author.

6. We note that Figures 3,7,9 and 10 in your submission contain [map/satellite] images which may be copyrighted. All PLOS content is published under the Creative Commons Attribution License (CC BY 4.0), which means that the manuscript, images, and Supporting Information files will be freely available online, and any third party is permitted to access, download, copy, distribute, and use these materials in any way, even commercially, with proper attribution. For these reasons, we cannot publish previously copyrighted maps or satellite images created using proprietary data, such as Google software (Google Maps, Street View, and Earth). For more information, see our copyright guidelines: http://journals.plos.org/plosone/s/licenses-and-copyright.

a. You may seek permission from the original copyright holder of Figure 3,7,9 and 10 to publish the content specifically under the CC BY 4.0 license.  

Reviewers' comments:

Reviewer's Responses to Questions

**Comments to the Author**

1. Is the manuscript technically sound, and do the data support the conclusions?

Reviewer #1: Yes

Reviewer #2: Yes

2. Has the statistical analysis been performed appropriately and rigorously? 

Reviewer #1: Yes

Reviewer #2: Yes

3. Have the authors made all data underlying the findings in their manuscript fully available?

Reviewer #1: Yes

Reviewer #2: Yes

4. Is the manuscript presented in an intelligible fashion and written in standard English?

Reviewer #1: Yes

Reviewer #2: Yes

5. Review Comments to the Author

Reviewer #1: This paper developed a framework for both quantitative and spatial node-based TOD measurement based on the four Ds and a line MRT 6 was selected as the study area. The topic is worth studying and fits within the journal scope, the experiments chosen and results highlighted are useful and illustrative. There are still some vague expressions in this paper, and some of which are not detailed enough. This is a pity, because I believe the scientific merit of the paper makes it worthy of publication. Therefore, I recommend some language editing to improve the readability of the paper. My specific comments are below:

1. Several related literatures are missing in the introduction.

DOI: 10.1016/j.ress.2021.108192

DOI: 10.1680/jmuen.18.00059

2. Page 11: In section 3.4, there are some TOD indicators such as ‘Population Density’. Please give the metrics for these indications. For example, the value of population density is within a certain range is very suitable for TOD.

3. Page 16, line 408: Please give why correlation analysis and sensitivity analysis are performed at the beginning of section 4.

4. Page 17, line 432: A lot of data are given in the tables, such as Table 3. Please give an example to explain the meaning of these data. The same is true for Table 4 and Table 5 in section 4.2.

5. Page 21, line 480: What is the null hypothesis?

6. Page 25, line 530: Please give an example to explain the meaning of figure 12.

7. Through the analysis of a series of TOD indicators in the manuscript, on the selected route MRT 6, compared with other stations, whether there are stations that are very suitable for TOD. Please give one or several such stations.

Reviewer #2: This study developed a framework for both quantitative and spatial node-based TOD measurement based on the four Ds (density, diversity, destination accessibility, and design) of the TOD concept. With 17 stations under construction, MRT 6 was selected as the study area. The TOD index was measured by nine indicators based on the four criteria (4Ds), spatially in the geographic information system (GIS). After calculating the indicators, the TOD index for each station’s 800m buffer was estimated using the spatial multi-criteria analysis (SMCA). The paper is generally well-written. The results are clearly explained, while the findings are discussed in a well-balanced manner. However, there are several issues that authors can work on:

1. Some contents should be reformulated. In the introduction section, it is more like "background - gap - contribution". I think the authors can first introduce TOD firstly, and then a gap, then what are you doing, and finally what are your contributions. In the last paragraph of the introduction section, the remainder of this paper organized should be given. Moreover, the first paragraph of the introduction is usually tightly focused on the topic, stating the necessity and importance of this study. The authors should clarify the research topic of this paper. Innovations and contributions are important. However, the authors do not clarify the innovations and contributions in the introduction section. Furthermore, the logic of the introduction section is loose, especially the connection between paragraphs.

2. On line 49, Page 2, what does “TODness” mean? The term is also mentioned several times in the article, and the authors should explain it further.

3. In the abstract section, the authors mentioned “developed a framework for both quantitative and spatial node-based TOD measurement”, but the quantitative analysis is not mentioned in the discussions and conclusions section.

4. The content and structure of Chapter 3 need to be adjusted. The study area and data collection should not be part of the methodology and should not be placed in the methodology section.

5. In the conclusion section, the authors mentioned that “sustainable development and (re)development policies should be implemented” Can you specify what these “policies” is, and can you give one or two examples to illustrate them?

6. Figure 12 is a bit hard to read and get the key points. Moreover, the authors do not analyze and discuss Figure 12, which seems to be only decorative in the text.

7. On line 269, Page 11, How was the “500-meter walking distance” obtained? and is there any reference to the corresponding literature? which should be further clarified in this study.

8. In Table 1, the scope of the study covers 2017. Is this not too late given that we are currently in August 2022? Can the data from five years ago now accurately provide useful information? Only17 stations have been chosen for node-based TOD index calculation, do the authors feel confident about the validity and generalizability of the findings?

9. There are some small details to modify.

a) Abbreviated words in the article should be given their corresponding full names when they first appear in the text. But the full names of MRT and BRT are not given in the article.

b) Text font and size should be consistent in all Figures. The font and size of Figure 6 are not consistent with the other Figures.

c) A table that spans pages should make a header for a continuation table on the second page, such as Table 3, on Page 18, and Table 6, on Page 28.

d) On line 248, Page 10, the format of literature citations should be consistent. “Balz and Schrijnen (2009) [48], Lindau et al. (2010) [49], and P. Newman (2009) [50]”, where “P. Newman (2009) [50]” is not consistent with other citations.

6. PLOS authors have the option to publish the peer review history of their article (what does this mean?). If published, this will include your full peer review and any attached files.

Reviewer #1: No

Reviewer #2: No

---

## [Author Response · Author response to Decision Letter 0]

17 Sep 2022

Reviewer #1

Comment-1:

Several related literatures are missing in the introduction.

DOI: 10.1016/j.ress.2021.108192

DOI: 10.1680/jmuen.18.00059

Reply to Comments:

The authors thank the reviewer for his/her comment. The authors have incorporated these two research papers in the introduction section. These have been incorporated as follows:

“For example, according to Liang et al. (2022) [5], bus priority on the road decreases traffic congestion, resulting in increased fuel efficiency and environmental preservation. In addition, a dedicated transit lane with an optimized signalized environment increases traffic circulation efficiency for other connecting modes. Moreover, location of transit node plays an important role on traffic circulation and commuter satisfaction. Passengers reap the most significant benefits when transit stops are strategically placed before and after the signalized intersections [6].”

Comment-2:

Page 11: In section 3.4, there are some TOD indicators such as ‘Population Density’. Please give the metrics for these indications. For example, the value of population density is within a certain range is very suitable for TOD.

Reply to Comments:

The authors thank the reviewer for his/her comment. The authors have incorporated these as follows:

“However, a reference value of PD is necessary for ideal TOD planning. According to the Florida Department of Transportation (FDOT) [58], PD of min 135 persons/acre, 100-145 persons/acre, and 80-135 persons/acre is ideal for urban core, urban general, and suburban areas respectively considering TOD.”

“The FDOT [58] recommends a commercial spaces-housing ratio of 10 commercial spaces per dwelling unit in urban cores and 5 commercial spaces per dwelling unit in other metropolitan regions.”

“Nevertheless, FDOT recommends the ED value of min 1000 jobs/acre, 190 to 250 jobs/acre and 35 to 80 jobs/acre for urban core, urban general and suburban areas respectively [58].”

“However, FDOT recommends 20/80, 50/50, 70/30 split between residential and non-residential land uses for urban core, urban general and suburban areas respectively [58].”

“However, FDOT recommends surface parking of maximum 10%, 70% and 80% of total area for urban core, urban general and suburban areas respectively [58].”

Comment-3:

Page 16, line 408: Please give why correlation analysis and sensitivity analysis are performed at the beginning of section 4.

Reply to Comments:

The authors thank the reviewer for his/her suggestion. The authors have incorporated these as follows:

“Firstly, a correlation analysis of the indicators has been conducted in this section. The correlation analysis will give insights into the relationship among the indicators, which will help planners and policymakers with proper TOD planning. Secondly, a sensitivity analysis has been executed. Typically, different scenarios are developed with different weightage in a sensitivity analysis. The reason for performing this analysis is that the SMCA is widely regarded as an effective tool for resolving spatial choice quandaries. However, its use of probable outcomes has been called into question [66]. In addition, some uncertainty is associated with the weights provided by those who lack extensive experience and knowledge [67]. So, from different scenarios with weightage, we can know the criteria of TOD changes to confirm how robust the model is. Then, a heatmap of the TOD index has been created, which will help the policymakers and urban-transport planners identify the high TOD zones for MRT 6. As a result, there will be alternative options for them to choose the situation better and implement it for a greater purpose. Moreover, a hotspot with spatial autocorrelation has been performed in this analysis for preliminary planning. These hotspot maps will be valuable for proper land use policymaking for the buffer of individual stations. Finally, the ranking of the stations based on the max TOD index has been performed to help policymakers to decide which buffer areas need more concern for TOD improvements.”

Comment-4:

Page 17, line 432: A lot of data are given in the tables, such as Table 3. Please give an example to explain the meaning of these data. The same is true for Table 4 and Table 5 in section 4.2.

Reply to Comments:

The authors thank the reviewer for his/her suggestion. The authors have incorporated these as follows:

“From Table 3, it has been found that the population density for all the stations ranges from 0 to 0.90. Moreover, for commercial and employment density, the standardized value ranges from 0 to 0.92 and 0.87. However, the maximum and minimum values for diversity are 0.54 and 1.0. However, maximum values for accessibility indicators such as mixedness, length of walkable/cyclable paths, and intersection density are 1.0, 0.91, and 0.87, respectively. Interestingly, there is zero parking space for almost all stations except Bijoy Sarani, Farmgate, Shahbag, BD Secretariat, and DU stations. The maximum value for open space is 1.0 for the maximum number of stations on MRT 6, and the minimum value is 0.11 for Uttara South station.”

“However, from Table 4, it has been found that scenario highest weightage is given to density and land use diversity (0.35) and minimum weightage is given to design (0.10) for base. Similarly, for scenario 2 and 3, maximum and minimum weightage is given to same as base scenario. But maximum weightage value is 0.40 in both cases. For scenario 3, maximum and minimum weightage is given to land use diversity (0.40) and design (0.1) respectively. It is noted that the weightage is given based on different ranking by modified rank-sum approach.”

“From Table 5, the maximum TOD index value for all 17 stations was observed within a reasonable range. We can observe that the highest deviation of the max value of the TOD index has ranged from 0.441 to 0.569. However, the max value varies for the stations. The max value ranges from 0.521 to 0.871 for the base scenario. However, for scenarios 1, 2, and 3, the maximum value range is 0.441 to 0.859, 0.473 to 872, and 0.561 to 0.871, respectively. Interestingly, the min value of the TOD index for all stations is 0. Moreover, the highest deviation of the mean value has ranged from 0.333 to 0.379, which is within the allowable limit. Nevertheless, the range of the mean value for the base scenario, scenarios 1, 2 and 3 is 0.172 to 0.429, 0.169 to 0.420, 0.175 to 0.256, 0.163 to 0.427, and 0.183 to 0.43. The range of the max, min, and mean values designated for the stations is identical for all scenarios. It is also to be noted that the stations’ overall ranking for all four scenarios has remained identical, indicating that the sensitivity analysis does not affect the results. Last but not least, the sensitivity analysis shows that the framework for figuring out index values is strong [55].”

Comment-5:

Page 21, line 480: What is the null hypothesis?

Reply to Comments:

The authors thank the reviewer for his/her suggestion. The necessary correction has been done as follows:

“In our autocorrelation, the null hypothesis is “the spatial distribution of the dataset is not clustered in nature”. However, in figure 8, it can be observed that the p-value is statistically significant, and the z-score is positive. Thus, the null hypothesis has been rejected. Therefore, the spatial distribution of high and low values of the dataset has been observed as more clustered than expected. So, underlying spatial processes are random [68].”

Comment-6:

Page 25, line 530: Please give an example to explain the meaning of figure 12.

Reply to Comments:

The authors thank the reviewer for his/her suggestion. The authors have incorporated these as follows:

“From Fig 12, it has been observed that the maximum value of LUM, CD, and OGS has been observed for the Shahbag station, which is the highest scoring station. Whereas Mirpur 10 and 11 depict almost the same indicator value except for CD, Mirpur 11 shows more value for CD. The maximum potential indicators for the Agargaon station are OGS, LUM, and LUD. However, CD, OGS, and LWC are suitable indicators for the Kazipara station. However, as the lowest scoring stations, DU, Uttara South, Shewrapara, Motijheel, and BD Secretariat stations have different indicator values. The highest indicators for DU stations are OGS and PU. The Uttara south, Shewrapara, and BD Secretariat stations show the highest value for LUM. Moreover, the Motijheel station offers the highest OGS value. However, medium scoring stations show variable indicator values. Therefore, individual station-based indicator analysis for the medium scoring stations will not be significant to discuss. So, based on the findings, the highest and lowest scoring stations are more important than the medium-scoring stations.”

Comment-7:

Through the analysis of a series of TOD indicators in the manuscript, on the selected route MRT 6, compared with other stations, whether there are stations that are very suitable for TOD. Please give one or several such stations.

Reply to Comments:

The authors thank the reviewer for his/her suggestion. These have been incorporated in discussion section as follows:

“Most TOD serving stations are Shahbag, Mirpur 11, Agargaon, Kazipara, and Mirpur 10, with high index scores from analyses of all scenarios. Max score ranges from 0.81 to 0.87. These stations are the beating hearts of Dhaka, showing the most significant potential. These max TOD degrees have been found in Dhaka's urban core stations. Typically, there is a consistent spatial pattern to the TOD degree in terms of its drop from the core to the periphery, regardless of city size [69]. However, to better plan, each station's buffer area, the indicators' values should also be considered. In addition, scores of the criteria such as land use diversity and density are greatly affected by the locations of the nodes, which means that the cost of current public transport services at these nodes is significantly higher than it is for other stations. 

Sahabag station has a relatively high score among all stations, considered one of Dhaka’s important transport hubs. The reason behind high values can be the highest level of retail facilities being available in this area in Dhaka. Second suitable TOD scoring stations is Mirpur 11. Commercial density is also highest for this station. So the reasons of highest TOD for these stations support the findings of Ganning and Miller (2020) [70] that the TOD Index appears to be the main driver of greater retail activity, with a correlation to a higher level of activity density. Other two suitable stations for TOD identified as Kazipara and Agargaon stations.”

Reviewer #2

Comment-1:

Some contents should be reformulated. In the introduction section, it is more like "background - gap - contribution". I think the authors can first introduce TOD firstly, and then a gap, then what are you doing, and finally what are your contributions. In the last paragraph of the introduction section, the remainder of this paper organized should be given. Moreover, the first paragraph of the introduction is usually tightly focused on the topic, stating the necessity and importance of this study. The authors should clarify the research topic of this paper. Innovations and contributions are important. However, the authors do not clarify the innovations and contributions in the introduction section. Furthermore, the logic of the introduction section is loose, especially the connection between paragraphs.

Reply to Comments:

The authors thank the reviewer for his/her comment. The necessary correction has been done and reformatted the introduction section as per the suggestions given.

Comment-2:

On line 49, Page 2, what does “TODness” mean? The term is also mentioned several times in the article, and the authors should explain it further.

Reply to Comments:

The authors thank the reviewer for his/her suggestion. These have been incorporated as follows:

“Evans and Pratt (2007) [8] highlighted that to evaluate the efficacy of TOD plans accurately, areas must also be assessed for their “TODness,” a quantitative measurement of TOD known as the TOD index. We can refer the term TODness as the degree of TOD. Zhou et al. (2019) [9] gave an elaborative definition of TODness. According to him, the magnitude to which the existing conditions of TOD sites fulfil accepted TOD principles frequently entailing significant expenditure and attention. The TOD principles include mixed and dense land use, accessibility, walking and cycling amenities, and compact development with pedestrian-friendly design. His definition also supports the idea of TODness given by Papa and Bertolini (2015) [10] and Singh et al. (2017) [7].”

Comment-3:

In the abstract section, the authors mentioned “developed a framework for both quantitative and spatial node-based TOD measurement”, but the quantitative analysis is not mentioned in the discussions and conclusions section.

Reply to Comments:

The authors thank the reviewer for his/her suggestion. In this paper, the authors do the quantitative measurement of TOD as TOD index value with other indicator-based analysis, such as correlation analysis of indicators, ranking of the stations, and sensitivity analysis have been included in the discussion section with their quantitative value. Moreover, other analyses such as heatmap, hotspot, and spatial autocorrelation have been done as spatial analyses of TOD. In conclusion, the quantitative value of TOD as the TOD index has been mentioned. So, to be more specific, the incorporated quantitively analysis has been incorporated as follows: 

“Most TOD serving stations are Shahbag, Mirpur 11, Agargaon, Kazipara, and Mirpur 10, with high index scores from analyses of all scenarios. Max score ranges from 0.81 to 0.87.”

“Sahabag station scoring 0.871 has a relatively high score among all stations, considered one of Dhaka’s important transport hubs. The reason behind high values can be the highest level of retail facilities being available in this area in Dhaka. Second suitable TOD scoring (0.836) stations is Mirpur 11. Commercial density (1.00) is also highest for this station.”

“Other two suitable stations for TOD identified as Kazipara (0.848) and Agargaon (0.844) stations.”

“For both, OGS indicator is common. LWC, LUM and LUD are also important indicators for these high scoring stations. So LUM, LWC, LUD and OGS are the indicators that should be given more importance for TOD planning in Dhaka. OGS correlates positively with LUD indicating that diversity in land use planning will be suitable for livable neighborhoods for the residents. Another critical indicator LUM shows an intense negative correlation with PD, which suggests that more people living near TOD stations have a less balanced land use mix. Therefore, residential land use should be balanced with other land uses. However, a negative correlation of LUM with design indicators indicates that balanced mixedness in land use promotes more walking, cycling, and a livable environment. So, destination accessibility criteria for the LUM indicator should be improved to get more captivity from TOD development. Moreover, according to Zacharias and Zhao (2018) [71] wide, straight pedestrian routes should be provided to the train stations to improve local accessibility with consideration of design criteria. So, for TOD planning in Dhaka as a developing city, a pedestrian and cyclist-friendly livable community design should be encouraged.”

“However, the minor scoring station’s index must be observed to identify the criteria for sustainable land use planning for TOD. DU (0.521), Uttara South (0.540), Shewrapara (0.677), BD Secretariat (0.703), Motijheel (0.709), and stations are the lowest-scoring stations. These neighborhoods are primarily residential. Not all nearby transportation nodes have parking areas that accommodate vehicles and non-motorized users.”

“Regarding the environment, including parks and open spaces, not all stations have been able to keep up. The lowest score has been found for the DU station. From indicator analysis, it has been found that less diversity value (0.00) is behind it. Only educational land use has been observed, indicating that diverse land use balanced with residential will be most suitable for TODness.”

“This paper has developed a framework for measuring TOD quantitatively (TOD index) around all transit nodes of MRT 6 in Dhaka using SMCA.”

Comment-4:

The content and structure of Chapter 3 need to be adjusted. The study area and data collection should not be part of the methodology and should not be placed in the methodology section.

Reply to Comments:

The authors thank the reviewer for his/her suggestion. The necessary correction has been done. In addition, the study and data collection have been designated as separate sections after the Methodology section.

Comment-5:

In the conclusion section, the authors mentioned that “sustainable development and (re)development policies should be implemented” Can you specify what these “policies” is, and can you give one or two examples to illustrate them?

Reply to Comments:

The authors thank the reviewer for his/her suggestion. These have been incorporated in conclusions section as follows:

“Regarding density and diversity, sustainable development and (re)development policies should be applied not only for MRT 6 but also for the Dhaka regions where TOD represents the city's future. However, some node, buffer area, and urban level development strategies can be implemented for long-term TOD planning. For example, the node-level policy includes preparing ahead of time for eki-naka development to prevent interrupting MRT routes and operations. Furthermore, buffer area level rules include modal transportation amenities, safety, comfort, and convenience for pedestrians, transit access, and park and ride. Finally, the incorporation of an official urban plan with the proposed MRT line, assimilation into the Detailed Area Plan (DAP) of the MRT district of Dhaka, and promotion of urban development along the transit corridor of Dhaka can be viable options for urban level development planning. Nevertheless, density zoning, reliable transportation infrastructure that fits new development trends, and land readjustment with acute phase development plans can all be recommended policies for (re)development.”

Comment-6:

Figure 12 is a bit hard to read and get the key points. Moreover, the authors do not analyze and discuss Figure 12, which seems to be only decorative in the text.

Reply to Comments:

The authors thank the reviewer for his/her suggestion. Updated figures have been added to make them clearer, and discussions have been incorporated for this as follows:

“From Fig 12, it has been observed that the maximum value of LUM, CD, and OGS has been observed for the Shahbag station, which is the highest scoring station. Whereas Mirpur 10 and 11 depict almost the same indicator value except for CD, Mirpur 11 shows more value for CD. The maximum potential indicators for the Agargaon station are OGS, LUM, and LUD. However, CD, OGS, and LWC are suitable indicators for the Kazipara station. However, as the lowest scoring stations, DU, Uttara South, Shewrapara, Motijheel, and BD Secretariat stations have different indicator values. The highest indicators for DU stations are OGS and PU. The Uttara south, Shewrapara, and BD Secretariat stations show the highest value for LUM. Moreover, the Motijheel station offers the highest OGS value. However, medium scoring stations show variable indicator values. Therefore, individual station-based indicator analysis for the medium scoring stations will not be significant to discuss. So, based on the findings, the highest and lowest scoring stations are more important than the medium-scoring stations.”

Comment-7:

On line 269, Page 11, How was the “500-meter walking distance” obtained? and is there any reference to the corresponding literature? which should be further clarified in this study.

Reply to Comments:

The authors thank the reviewer for his/her suggestion. These have been incorporated as follows:

“Furthermore, unlike traditional TOD developments, specific TOD areas have been covered by a 500-meter walking distance. The National Transit Oriented Development (TOD) Policy defines the influence zone of TOD as 500-800 m if the station spacing is approximately 1 km. If the station spacing is less than 1 km, the influence zone becomes 500 m because of overlapping [56]. Moreover, according to TOD standard framework by Institute for Transit Development & Policy (ITDP), within a 5 km radius, rapid transit connections with frequent bus stops should be accessible at around 500 m. However, as our study is for a developing south Asian city, we take the guideline of Cochin and Mumbai city metro by considering various south Asian city’s guidelines. The influence area for these two cities has been adopted as a 500 m buffer for Both cities [57]. As both cities are densely populated, like Dhaka, we take their guideline for our analysis radius.”

Comment-8:

In Table 1, the scope of the study covers 2017. Is this not too late given that we are currently in August 2022? Can the data from five years ago now accurately provide useful information? Only17 stations have been chosen for node-based TOD index calculation, do the authors feel confident about the validity and generalizability of the findings?

Reply to Comments:

The authors thank the reviewer for his/her suggestion. Using the data of 2017 can provide useful information about the research and the authors are confident about the validity and generalizability of the findings for four reasons:

1. Dhaka is a developing south Asian city; in a developing city, data is very scarce and unreliable, most importantly, transportation-related data. So, in 2017 data that has been collected from World Bank, which is the most reliable source. Unfortunately, the last update of this data for Dhaka was in 2017. Therefore, sthe authors have to use it for reliability purposes. Moreover, other available sources are not reliable.

2. No TOD index study has been conducted for Dhaka before. Most importantly, without the TOD index study, the MRT 6 has gone to the implementation stage in 2016. Here, the authors try to find the gap. They try to make a framework for a developing city so that in the future, others can follow this framework. For the validity of the findings, sensitivity analysis has been conducted. So, the authors try to make a case study for MRT 6, and these findings are especially for MRT 6 future improvements and planning. Based on the results, this will help other TOD plans in Dhaka city in the future. So, this study is an aiding tool for planners and policymakers.

3. Dhaka is a highly densely developed city. As the stations of MRT 6 are situated primarily in the urban core areas, the areas are already highly developed. So, the land use, building footprint, and road network have not changed significantly from 2017 to 2022. Thus, the authors are confident about providing useful information from the collected data. 

4. As MRT 6 went to the implementation stage in 2016, timeframe of our collected data (2017) is after the implementation time of MRT 6. So, it is logical to the more accurate information of the TOD index during the implantation stage scenario for the built area of the stations of MRT 6.

Comment-9:

9. There are some small details to modify.

a) Abbreviated words in the article should be given their corresponding full names when they first appear in the text. But the full names of MRT and BRT are not given in the article.

b) Text font and size should be consistent in all Figures. The font and size of Figure 6 are not consistent with the other Figures.

c) A table that spans pages should make a header for a continuation table on the second page, such as Table 3, on Page 18, and Table 6, on Page 28.

d) On line 248, Page 10, the format of literature citations should be consistent. “Balz and Schrijnen (2009) [48], Lindau et al. (2010) [49], and P. Newman (2009) [50]”, where “P. Newman (2009) [50]” is not consistent with other citations.

Reply to Comments:

The authors thank the reviewer for his/her suggestion. The necessary corrections has been done and incorporated as below:

a) “Five mass rapid transit (MRT) lines and two bus rapid transit (BRT) lines have been 

proposed for Dhaka, with MRT 6 and BRT 7 currently under construction [17].”

b) Corrected as per the comments.

c) Corrected as per the comments.

d) “Furthermore, several well-known and widely used indicator variables used in several successful TOD case studies have been identified by Balz and Schrijnen (2009) [51], Lindau et al. (2010) [52], and Newman (2009) [53].”

---

## [Decision Letter · Decision Letter 1]

26 Dec 2022

A framework to measure transit-oriented development around transit nodes: Case study of a mass rapid transit system in Dhaka, Bangladesh

PONE-D-22-18746R1

Dear Dr. Anwar,

We’re pleased to inform you that your manuscript has been judged scientifically suitable for publication and will be formally accepted for publication once it meets all outstanding technical requirements.

Kind regards,

Sheng Jin

Academic Editor

PLOS ONE

Additional Editor Comments (optional):

Reviewers' comments:

Reviewer's Responses to Questions

**Comments to the Author**

1. If the authors have adequately addressed your comments raised in a previous round of review and you feel that this manuscript is now acceptable for publication, you may indicate that here to bypass the “Comments to the Author” section, enter your conflict of interest statement in the “Confidential to Editor” section, and submit your "Accept" recommendation.

Reviewer #1: All comments have been addressed

2. Is the manuscript technically sound, and do the data support the conclusions?

Reviewer #1: Yes

3. Has the statistical analysis been performed appropriately and rigorously? 

Reviewer #1: Yes

4. Have the authors made all data underlying the findings in their manuscript fully available?

Reviewer #1: Yes

5. Is the manuscript presented in an intelligible fashion and written in standard English?

Reviewer #1: Yes

6. Review Comments to the Author

Reviewer #1: All the issues mentioned in first round review have been addressed. The paper can be published in current format.

7. PLOS authors have the option to publish the peer review history of their article (what does this mean?). If published, this will include your full peer review and any attached files.

Reviewer #1: No

---

## [Editor Report · Acceptance letter]

29 Dec 2022

PONE-D-22-18746R1 

A framework to measure transit-oriented development around transit nodes: Case study of a mass rapid transit system in Dhaka, Bangladesh 

Dear Dr. Uddin:

I'm pleased to inform you that your manuscript has been deemed suitable for publication in PLOS ONE. Congratulations! Your manuscript is now with our production department. 

Kind regards, 

on behalf of

Dr. Sheng Jin 

Academic Editor

PLOS ONE